# DIMENSION DOMAIN CO-DECOMPOSITION: SOLVING PDES WITH INTERPRETABILITY

## Abstract

Physics-informed neural networks (PINNs) have demonstrated effectiveness in solving partial differential equations (PDEs), yet they often struggle in high-dimensional regimes and lack interpretable representations and in scenarios involving sharp solution structures. Moreover, existing approaches typically rely on manually specified domain partitions. We propose a unified Dimension–Domain Co-Decomposition (3D) framework that jointly integrates dimension-wise decomposition with mixture-of-experts (MoE)–based domain decomposition. At the dimension level, we introduce an interpretable decomposition mechanism in which coordinate inputs are decoupled within each expert through a shared MLP with indexed inputs, enabling parameter efficiency while preserving expressivity. To quantitatively assess interpretability, we define a Variable Interpretability (VI) metric that measures the alignment between learned latent components and the corresponding solution factors. At the domain level, an MoE-based gating mechanism adaptively partitions the solution space without requiring predefined regions or interface conditions. Extensive experiments on PDE benchmarks demonstrate that the proposed framework achieves improved accuracy and computational efficiency compared to standard PINNs and related baselines, while providing interpretable and scalable representations.

## 1 Introduction

Partial differential equations (PDEs) provide the mathematical foundation for describing a wide range of physical and engineering phenomena, including fluid dynamics Anderson (1995), wave propagation Strauss (2007), and quantum mechanics Griffiths & Schroeter (2018). Classical numerical solvers such as the finite element method (FEM) Zienkiewicz et al. (2005); Babuška (1971), the finite difference method (FDM) LeVeque (2007); Lax & Richtmyer (1956), and the spectral method (SM) Trefethen (2000); Boyd (2001) have long been the standard tools for approximating PDE solutions. FEM is flexible for handling irregular domains, FDM is simple and efficient on structured grids, while classical SM achieves exponential convergence, they rely on dense grid constructions. Despite their success, both methods suffer from rapidly increasing computational cost when dealing with high-dimensional problems, complex nonlinearities, or solutions with sharp local features, which often makes them impractical for large-scale applications.

In recent years, neural networks have emerged as promising alternatives for PDE solving, either by directly approximating solutions from data or by embedding the governing equations into the training objective through physics-informed neural networks (PINNs) Raissi et al. (2019). PINNs offer advantages in high-dimensional settings where mesh-based numerical solvers become computationally intractable. Though classical sparse grid methods (Smolyak, 1963) alleviate the curse of dimensionality to some extent and modern Tensor-Train (TT) decompositions (Richter et al., 2021) bypass the dimensional bottleneck by exploiting low-rank tensor representations, PINNs offer an alternative, highly flexible mesh-free approach. Intriguingly, the core philosophy of the TT format shares a profound mathematical intuition with our dimension decomposition architecture. Building on this flexibility, two major lines of decomposition-based methods have been explored to further enhance scalability and adaptivity. Dimension decomposition improves scalability by factorizing solutions along coordinates Cho et al. (2023); Liu et al. (2024). This strategy simplifies optimization in high-dimensional settings and further mitigates the curse of dimensionality. However, existing approaches

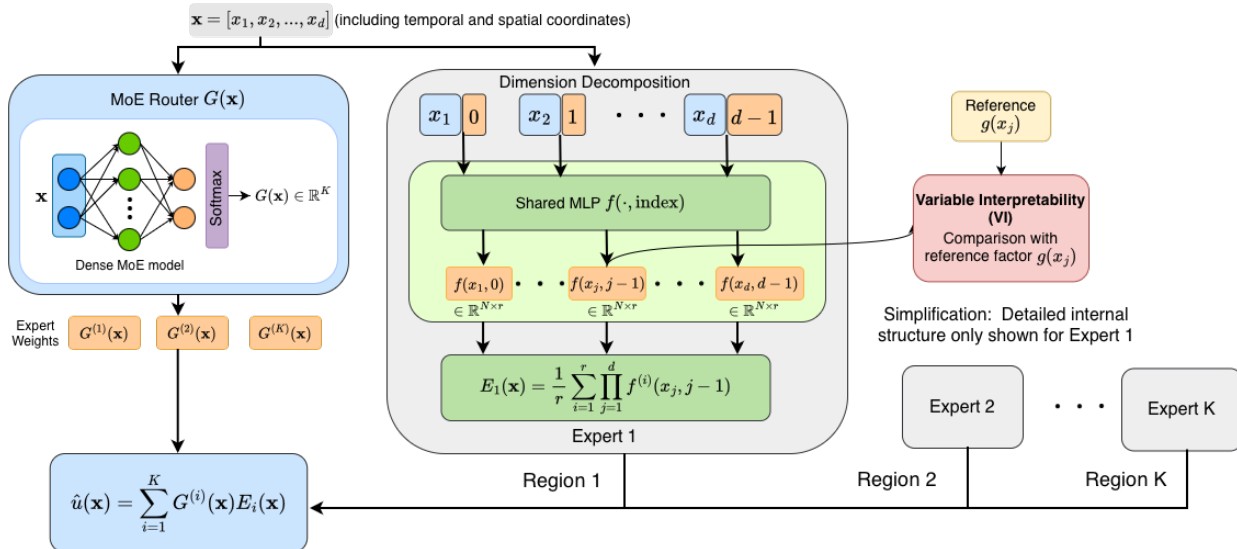

Figure 1: **Overview of the proposed Dimension–Domain Co-Decomposition (3D) framework.** Given an input $x = [x_1, \ldots, x_d]$, a Mixture-of-Experts (MoE) router $G(x)$ produces adaptive gating weights over $K$ experts. Within each expert, a dimension-wise decomposition is performed: individual coordinate inputs are processed through a shared MLP $f(\cdot, \text{index})$ with indexed inputs, generating per-dimension latent components. These components are aggregated to form the expert output $E_i(x)$. The architecture implicitly induces domain decomposition of the input space via the routing mechanism. The resulting regions can be overlapping, as multiple experts may be simultaneously activated for a given input. The final prediction is computed as a weighted combination $\hat{u}(x) = \sum_{i=1}^{K} G^{(i)}(x) E_i(x)$. Variable Interpretability (VI) measures the alignment between each learned dimension-specific component and the corresponding reference factor, providing a quantitative assessment of interpretability.

lack interpretability measurement. In contrast, domain decomposition Jagtap et al. (2020a); Shukla et al. (2021); Hu et al. (2023) focuses on local adaptivity by dividing the computational domain into smaller subdomains, with each subdomain handled by a specialized model. This enables better approximation of both smooth and discontinuous solutions. Nevertheless, such methods typically rely on manually pre-defining the subdomains. When the subdomains overlap, one must introduce extra loss terms to ensure the predictions agree in the overlapping regions; when the subdomains are non-overlapping, additional conditions are required to enforce continuity across the shared boundaries. These constraints make the training procedure more complicated and problem-dependent.

To overcome these limitations, we propose Dimension Domain Co-Decomposition (3D), a unified framework that combines both decomposition strategies in a scalable, interpretable, and fully automatic manner. At the dimension level, each variable is modeled separately, which improves scalability in high dimensions. In practice, these dimension components are processed through a shared MLP, ensuring parameter efficiency across coordinates. At the domain level, 3D employs Mixture of Experts (MoE) Jacobs et al. (1991). It contains multiple experts, and a router assigning soft weights to combine their outputs. This mechanism encourages each expert to specialize in certain subregions, so that domain decomposition emerges adaptively during training. As a result, 3D can effectively capture solutions with sharp local features without requiring pre-defined regions or explicit interface conditions. An illustration of 3D with K experts on input $\mathbf{x} \in \mathbb{R}^d$ is shown in Figure 1. In addition, we propose Variable Interpretability (VI), a quantitative metric that matches predicted per-dimension components to ground-truth factors.

We summarize our contribution as follows:

- **Dimension Domain Co-Decomposition:** We propose 3D, a unified framework that simultaneously (i) factorizes the solution along input dimensions via a low-rank separable parameterization and

(ii) induces adaptive domain decomposition via a dense Mixture-of-Experts (MoE) router, enabling accurate learning of sharp-featured PDE solutions without manual subdomain design or interface penalties.

- **Shared-MLP dimension decomposition with index encoding:** Within each expert, we introduce a single shared MLP that takes coordinate value and dimension index as input and outputs rank-$r$ factors, achieving dimension-wise specialization with parameter sharing and making the per-expert parameter count independent of the PDE dimension $d$, while preserving the expressivity of per-dimension subnetworks in the large-width regime.

- **A quantitative interpretability metric with invariances:** We introduce Variable Interpretability (VI), a novel and scale-invariant metric based on principal angles between subspaces, to quantitatively evaluate whether learned dimension factors recover ground-truth components, moving beyond qualitative factor visualization.

- **Empirical evidence:** Across high-dimensional and sharp-interface PDE benchmarks, we show that 3D improves accuracy and training stability, achieves large parameter reductions via sharing, and learns physically aligned gating partitions (e.g., shocks/interfaces) without handcrafted decomposition rules.

## 2 Related Work

### 2.1 Dimension Decomposition and Interpretability

High-dimensional PDEs pose significant challenges for neural network-based solvers. Building on the PINNs framework, several recent works Cho et al. (2023); Liu et al. (2024); Vemuri et al. (2024); Liu et al. (2022) introduce dimension-wise decomposition strategies to mitigate the curse of dimensionality. Most of these approaches rely on classical tensor decomposition techniques Tucker (1966); Carroll & Chang (1970), which improve efficiency by reducing the representation complexity, but still assign a separate neural network to each dimension, leading to suboptimal efficiency. At the same time, these methods offer little interpretability measurements of the learned components. In parallel, the interpretable machine learning community has developed models such as GAMs, NAMs, and self-explaining networks Hastie & Tibshirani (1990); Wood (2017); Agarwal et al. (2021); Alvarez-Melis & Jaakkola (2018); Lou et al. (2013), which represent the target function as a sum of univariate functions, each depending on a single variable. These models offer intuitive per-variable explanations, but their additive structure struggles to capture higher-order interactions, which are often intrinsic to PDE solutions. Beyond additive models, sparse regression-based methods such as SINDy and its variants Brunton et al. (2016); Kaiser et al. (2018) provide another line of interpretability by discovering governing equations from data. Unlike variable-wise interpretability, these methods explain the underlying physical laws by identifying symbolic equations, rather than uncovering the structures of PDE solutions themselves. To fill in these gaps, we propose a shared-MLP dimension decomposition that removes redundant per-dimension networks for greater efficiency, and introduce Variable Interpretability (VI), a metric quantifying the alignment between learned components and ground-truth factors.

### 2.2 Domain Decomposition of PINNs

Domain decomposition has been widely adopted to improve PINNs for solving complex PDEs. The XPINNs framework Jagtap et al. (2020c) pioneered this idea by partitioning the computational domain into multiple subdomains and training a separate PINN in each region; to ensure consistency, XPINNs enforces continuity of the solution across subdomain interfaces through additional interface losses. Subsequent works have refined this approach: Shukla et al. Shukla et al. (2021) introduced parallel implementations combining cPINNs Jagtap et al. (2020b) and XPINNs, exploiting overlapping Schwarz-type decompositions to better handle multi-scale problems. Hu et al. (2023) proposed APINNs, which use soft gating mechanisms to allow more flexible domain decomposition. Dolean et al. Dolean et al. (2024) developed multilevel decomposition architectures to improve accuracy for large or highly heterogeneous domains. More recently, the approach

named BPINN Vicens Figueres et al. (2025) integrates Bayesian PINNs with domain decomposition, computing local uncertainties concurrently and enforcing interface flux continuity among subdomains. There are also specialized applications, such as domain decomposition PINNs for incompressible Navier–Stokes equations Gu et al. (2024). Despite these advances, a common limitation is that all existing approaches require predefined partitions of the computational domain. Moreover, additional conditions must be imposed at the subdomain interfaces to guarantee continuity of the solution. These requirements restrict adaptivity and limit the flexibility of domain decomposition when applied to PDEs with unknown or heterogeneous solution structures. In contrast, our framework enables automatic and adaptive domain decomposition during training.

# 3 Dimension Domain Co-Decomposition

Existing PINNs-based methods for high-dimensional PDEs suffer from three obstacles: (i) high computational cost due to the need for dense collocation sampling; (ii) a lack of principled interpretability metric for dimension-wise factorizations, where scaling, permutation, and cross-dimension mixing obscure whether learned components reflect the underlying physics; and (iii) brittle domain decomposition that depends on predefined subdomains and delicately tuned interface penalties, making performance sensitive to the chosen partition and enforcement strength. To address these issues, we adopt a dimension decomposition that reduces computation-graph complexity by combining them in a low-rank manner; we introduce Variable Interpretability (VI) to quantify alignment between learned per-dimension components and reference factors; and we develop MoE-driven domain decomposition that maps the input coordinates to soft expert assignments, avoiding manual region design and explicit interface enforcement. In combination, the dimension decomposition lowers training cost, VI provides quantitative interpretability, and the MoE router delivers robust, automatic domain partitioning. Given input $\mathbf{x} = [x_1, x_2, \cdots, x_d]$, the predicted solution $\hat{u}$ takes the form:

$$\hat{u}(\mathbf{x}) = \sum_{i=1}^{K} G^{(i)}(\mathbf{x}) E_i(\mathbf{x}),$$

$$E_i(\mathbf{x}) = E_i\big(f_1(x_1), f_2(x_2), \cdots, f_d(x_d)\big)$$
(3.1)

where $f_j$ for $j = 1, 2, \cdots, d$ stands for the Multilayer Perceptron (MLP) processing each dimension component. $E_i$ for $i = 1, 2, \cdots, K$ represents expert while $G(\mathbf{x}) \in \mathbb{R}^K$ is a router assigning weights for experts. The model remains differentiable with respect to the input and all network parameters. In particular, gradients propagate through the entire mixture structure, i.e., both the router and the experts. Consequently, PDE residuals backpropagate through the weighted summation, leading to the joint optimization of the routing and experts networks.

## 3.1 Dimension Decomposition in 3D Framework

Conventional methods mix all dimensions in a single network. For high-dimension problems, large number of data complicates the computation graph, making both forward and, more severely, backward propagation expensive. We adopt dimension decomposition in single expert to decouple coordinates and simplify both forward propagation and derivative computation. Our domain decomposition design is similar in form to the Canonical Polyadic Decomposition (CP-decomposition) Carroll & Chang (1970); Harshman (1970). Conventionally, for $d$-dimensional input, the output can be written as follows:

$$\hat{u}(x_1, \ldots, x_d) = \frac{1}{r} \sum_{i=1}^{r} \prod_{j=1}^{d} f_j^{(i)}(x_j)$$
(3.2)

where $\hat{u} : \mathbb{R}^d \to \mathbb{R}$ is the predicted solution, $x_j \in \mathbb{R}$ is a coordinate of $j$-th component including temporal coordinates if exist. To ensure a fair comparison across different ranks, we introduce a scaling factor $\frac{1}{r}$ to normalize the summation. This design ensures that the magnitude of $\hat{u}$ remains invariant to the choice of $r$, thereby facilitating stable training and consistent hyperparameter tuning (e.g., the loss weights $w_{pde}, w_{ic}, w_{bc}$

from equation 3.4) across different ranks. $f_j(x_j) : \mathbb{R} \to \mathbb{R}^r$ represents independent MLP processing $x_j$. $r$ is comparable to the rank in CP-decomposition. In our settings, $r$ impacts more on Variable Interpretability (VI) than accuracy. Modest $r$ are sufficient-typically $r \in \{4, \cdots, 16\}$ achieving good interpretability while maintaining satisfactory accuracy, see section 4.

However, independent per-axis processing introduces a large number of parameters. We address this issue by using a single shared MLP to model all dimension components within each expert. Specifically, each component is represented by a two-dimensional input vector consisting of the coordinate value and its index. For the $j$-th dimension component, the corresponding output is given by $f(x_j, j-1)$. For example, for 3d PDE problem, outputs of dimension components are $f(x_1, 0), f(x_2, 1), f(x_3, 2)$. We treat temporal coordinate $t$ as part of the physical vector coordinates. Therefore, equation 3.2 can be rewritten into:

$$\hat{u}(x_1, \ldots, x_d) = \frac{1}{r} \sum_{i=1}^{r} \prod_{j=1}^{d} f^{(i)}(x_j, j-1) \tag{3.3}$$

The shared formulation introduces an implicit coupling across dimensions at the representation level, which acts as an inductive bias. To further illustrate the effectiveness of the proposed method, we provide a universal approximation theorem and prove it in Appendix A.

**Theorem 1** (Universal Approximation with Shared MLP). *Let $\Omega \subset \mathbb{R}^d$ be compact and let $u \in L^2(\Omega)$. Then for any $\varepsilon > 0$, there exists an integer $r > 0$ and a neural network $f : \mathbb{R} \times \{0, 1, \ldots, d-1\} \to \mathbb{R}^r$ such that*

$$\hat{u}(\mathbf{x}) = \frac{1}{r} \sum_{i=1}^{r} \prod_{j=1}^{d} f^{(i)}(x_j, j-1)$$

*satisfies*

$$\|u - \hat{u}\|_{L^2(\Omega)} < \varepsilon.$$

By the above Theorem 1, we demonstrate that any functions in $L^2$ space can be approximated by the proposed shared MLP architecture.

Our framework is based on PINNs. Therefore, the loss function can be written as follows:

$$Loss = w_{pde}Loss_{pde} + w_{ic}Loss_{ic} + w_{bc}Loss_{bc} \tag{3.4}$$

where $Loss_{pde}$ is the PDE residual loss, which penalizes the discrepancy between the neural network prediction substituted into the PDE and the equation's right-hand side at sampled collocation points. $Loss_{ic}$ and $Loss_{bc}$ represent initial-condition loss (for time dependent problems) and boundary-condition loss, respectively.

The proposed architecture is related to Separable Physics-Informed Neural Networks (SPINNs) (Cho et al., 2023), but it departs from them in several key aspects that yield significant advantages. First, we use single MLP processing each dimension component with an embedded index as input, decreasing the trainable parameters when handling high-dimensional problems, see section 4 for more information. Second, our framework naturally integrates with a MoE structure. While SPINNs rely on forward-mode automatic differentiation (AD), this is inherently incompatible with MoE because the router breaks the separable structure. Instead, we adopt reverse-mode AD which allows the decomposition to remain effective while benefiting from adaptive domain specialization. Furthermore, the introduction of a scaling factor $\frac{1}{r}$ ensures that the output of each individual expert remains numerically stable regardless of the rank $r$. This normalization subsequently propagates to the final MoE prediction, preventing magnitude shifts that could otherwise destabilize the training of the router and the overall weighted summation. Lastly, SPINNs are constrained to rectangular tensor-product grids, our dimension decomposition design operates on mesh-free collocation points. Instead of constructing a restrictive axis-aligned grid, we independently sample training points for each dimension and then combine them in a use to retain maximum geometric flexibility while maintaining training efficiency.

## 3.2 Variable Interpretability (VI)

Existing dimension decomposition techniques often lack quantitative tools to evaluate the quality of the learned components. To address this, we introduce the *Variable Interpretability* (VI) metric, which measures the geometric alignment between the learned functional subspace and the reference target (derived from analytical solutions). This metric is theoretically grounded in the concept of *Principal Angles* between subspaces.

Let $\mathcal{D}_j = \{x_j^{(1)}, \ldots, x_j^{(N)}\}$ denote a set of $N$ sample points for the $j$-th dimension. We construct the learned feature matrix $\mathbf{F}_j \in \mathbb{R}^{N \times r}$ and the target matrix $\mathbf{T}_j \in \mathbb{R}^{N \times s}$ by evaluating the expert outputs and the reference solution on $\mathcal{D}_j$, respectively. Here, $s$ represents the rank of the ground truth (typically $s = 1$ for separable analytical solutions), while $r$ is the rank of our decomposition model. For brevity, we omit the subscript $j$ in the following derivation.

**Preprocessing.** To ensure scale and shift invariance, we first center and normalize the columns of both matrices. Let $\mathbf{f}_k$ be the $k$-th column of $\mathbf{F}$. The normalized matrix $\bar{\mathbf{F}}$ is constructed by transforming each column:

$$\bar{\mathbf{f}}_k = \frac{\mathbf{f}_k - \mu_k \mathbf{1}}{\|\mathbf{f}_k - \mu_k \mathbf{1}\|_2 + \epsilon}, \quad k = 1, \ldots, r \tag{3.5}$$

where $\mu_k$ is the mean of $\mathbf{f}_k$, $\mathbf{1} \in \mathbb{R}^N$ is a vector of ones, and $\epsilon$ is a small constant for numerical stability. While the subsequent principal angles are scale-invariant, rescaling is introduced to enhance numerical stability. The target matrix $\mathbf{T}$ is processed similarly to obtain $\bar{\mathbf{T}}$.

**Subspace Alignment.** We perform reduced QR decomposition to obtain the orthonormal bases for the spans of the normalized matrices:

$$\mathbf{Q}_F \mathbf{R}_F = \bar{\mathbf{F}}, \quad \mathbf{Q}_T \mathbf{R}_T = \bar{\mathbf{T}}, \tag{3.6}$$

where $\mathbf{Q}_F \in \mathbb{R}^{N \times r'}$ and $\mathbf{Q}_T \in \mathbb{R}^{N \times s'}$ are matrices with orthonormal columns. We then compute the singular values $\{\sigma_i\}_{i=1}^m$ of the matrix product $\mathbf{Q}_F^\top \mathbf{Q}_T$, where $m = \min(r', s')$. These singular values represent the cosines of the principal angles between the learned and target subspaces.

**Definition.** The VI for the $j$-th dimension is defined as the mean squared cosine similarity:

$$VI_j = \frac{1}{m} \sum_{i=1}^m \sigma_i^2. \tag{3.7}$$

VI is calculated within each expert for each dimension component. The score $VI_j \in [0, 1]$ serves as a scalar proxy for subspace similarity. The score $VI_j \in [0, 1]$ serves as a scalar proxy for the alignment between the predicted and target subspaces. A value close to 1 indicates that the subspace spanned by the learned expert components effectively captures the essential functional variations of the ground truth.

We emphasize that the geometric alignment captured by VI is functionally decoupled from a well-learned global solution. An ill-trained network can match a subspace feature with constant bias, leading to high VI with high error as well. Therefore, VI is conditioned on good learning performance. It is an orthogonal validation metric that becomes physically meaningful only after the model converges to the correct global solution manifold.

Notably, this metric evaluates the representation features as a collective whole, assessing the alignment and containment of the subspace spanned by $\mathbf{Q}_T$ within the subspace spanned by $\mathbf{Q}_F$. In practical scenarios, the target matrix $\mathbf{T}$ often has a shape of $(N, s)$ with $s \leq r$, where $r$ is the chosen decomposition rank. Consequently, the dimensionality of the exact basis may be lower than that of the predicted basis.

When $s < r$, a score of $VI = 1$ implies that the ground-truth subspace is *fully contained* within the predicted subspace, rather than requiring the two spaces to be identical. This property is particularly relevant for over-parameterized regimes where $r$ is chosen to be sufficiently large to capture the underlying physics, while the number of intrinsic basis vectors $s$ remains small. In the special case where $s = r$, $VI = 1$ indicates that the

predicted and exact subspaces are identical. For instance, in the 5D Poisson equation discussed in Section 4, the exact $x$-component yields $s = 1$, while we typically employ $r > 1$; here, $VI \approx 1$ confirms that the predicted components effectively recover the required analytical basis.

*Remark* 1 (Degenerate Case). Consider a scenario where the ground-truth factor is a constant function (e.g. $u(x, y) = 5\sin(y)$). After the mean removal, the corresponding shifted columns degenerates into zero, yielding $VI_x = 0$. Physically, this implies $x$-component carries no information to the final solution.

### 3.3 MoE-Driven Domain Decomposition

Partitioning the solution domain into subdomains enables local specialization of the underlying physics, improving accuracy and stability. Previous domain decomposition methods require manually pre-defined regions and interface conditions. To achieve automatic and adaptive domain decomposition, we adopt a Dense MoE model (Jacobs et al., 1991). Compared with Sparse MoE (Shazeer et al., 2017), dense MoE avoids expert collapse and provides more stable training. This is important in problems with shocks where top-$k$ gating may cause instability near shocks. Router is a MLP $G : \mathbb{R}^d \to \mathbb{R}^K$ taking only $\mathbf{x} \in \mathbb{R}^d$ (including temporal and spatial coordinates) as input. It produces logits which are then converted into mixture weights via a softmax. The weight assignment serves as a soft partition indicator – large weight marks the region where expert is responsible for. Each expert $E_i$ for $i = 1, 2, \cdots, K$ specializes in local regions. It remains smooth within its responsible region while differing from other experts to cover complementary behaviors. Together, they provide global approximation by $\sum_{i=1}^{K} G^{(i)}(\mathbf{x})E_i(\mathbf{x})$. The above Theorem 1 characterizes the approximation capability of a single expert. Since the MoE architecture forms a convex combination of expert outputs, it does not reduce the expressive power and can only further enhance the representation capacity.

Since the predicted solution is the weighted sum of experts' outputs, the overall loss function follows equation 3.4, except for the computation of $\hat{u}$. All experts share same architectures and inputs with separate parameters. End-to-end training is performed. Both the router and experts are updated via gradient descent optimization. Our experiment results demonstrate that increasing the number of experts $K$ initially leads to significant error reduction and reflects finer domain decomposition. However, beyond a certain number $K_{optimal}$, additional experts yield similar errors and no more new information about domain decomposition. In practice, we determine $K_{optimal}$ via ablation studies by progressively increasing $K$ and selecting the smallest value beyond which additional experts provide negligible error reduction.

## 4 Experiment

### 4.1 Experimental Setup

We evaluate the performance and interpretability of the proposed framework across several benchmark PDEs. The experiments are categorized into two primary investigative tracks:

1. **Dimension Decomposition**: Focused on the Poisson and Wave equations to validate the core dimension decomposition module with shared MLP.

2. **MoE-driven Domain Decomposition**: Applied to non-linear systems including the Viscous Burgers, Advection-Diffusion-Reaction (ADR), Allen-Cahn equation and another 5d Poisson case, where each expert incorporates the shared dimension decomposition architecture.

Detailed formulations and specific parameter settings for all PDEs are provided in the Appendix B.

**Network Configuration and Routing.** Our framework utilizes a unified expert design where each expert employs a shared MLP architecture for dimension decomposition. This architecture can either operate as a single module (e.g., for Poisson and Wave equations) or be integrated into a Mixture-of-Experts (MoE) structure. For domain decomposition tasks, we implement a dense MoE with a router consisting of a 5-layer MLP. Each layer in the router has a width of 64 neurons and utilizes the Tanh activation function.

Table 1: Performance across PDE benchmarks. Relative $\ell_2$ errors ($\times 10^{-4}$) are reported as mean $\pm$ std over 5 random seeds. Time denotes total training time in seconds.

| Problem | $r$ | $\ell_2$ Error | Time (s) |
|---|---|---|---|
| Poisson (5d) | 1 | $1.93 \pm 0.94$ | 576.81 |
| Poisson (10d) | 5 | $13.79 \pm 6.35$ | 1839.03 |
| Wave 1d ($c = 2$) | 1 | $0.09 \pm 0.04$ | 174.05 |
| Wave 1d ($c = 5$) | 16 | $259.49 \pm 256.20$ | 336.49 |
| Wave 2d ($c = 2$) | 1 | $5.96 \pm 2.56$ | 460.60 |
| Viscous Burgers | 16 | $5.68 \pm 2.20$ | 399.67 |
| ADR (Traveling Wave) | 2 | $17.13 \pm 10.45$ | 366.79 |
| ADR (Oscillatory Decay) | 4 | $4.99 \pm 2.44$ | 430.86 |
| Allen–Cahn | 16 | $155.05 \pm 123.38$ | 1108.77 |
| 5d nonlinear Poisson | 5 | $6.45 \pm 1.78$ | 1358.38 |

Table 2: Comparison of trainable parameters between unshared MLPs and shared MLP ($r = 16$).

| Problem | $K$ | Unshared MLPs | Shared MLP |
|---|---|---|---|
| Poisson (5d) | 1 | 26640 | **5392** |
| Poisson (10d) | 1 | 53280 | **5392** |
| Wave 1d | 1 | 10656 | **5392** |
| Wave 2d | 1 | 15984 | **5392** |
| Viscous Burgers | 2 | 34114 | **23586** |
| ADR (Traveling Wave) | 3 | 44835 | **29043** |
| ADR (Oscillatory Decay) | 4 | 55556 | **34500** |
| Allen–Cahn | 2 | 34114 | **23586** |
| 5d nonlinear Poisson | 2 | 34114 | **23586** |

**Training Strategy and Implementation.** The training process is conducted in two stages: an initial phase using the Adam optimizer to ensure fast convergence, followed by L-BFGS for local refinement and higher precision. A cosine-annealing scheduler is employed to adaptively adjust the learning rate during the Adam phase. All models are implemented in PyTorch and trained on a heterogeneous hardware environment including NVIDIA RTX 5090 and NVIDIA L40 GPUs. All training time comparisons are measured on NVIDIA L40 GPUs to ensure fair and consistent evaluation. Performance is quantified using the relative $\ell_2$ error.

## 4.2 Performance Overview

In this section, we present a high-level summary of our framework's performance across various PDE benchmarks. Table 1 reports the relative $\ell_2$ errors and total training times. As observed, our method consistently achieves high-accuracy approximations, with most errors maintained at the $10^{-3}$ to $10^{-4}$ level. Significantly, the framework demonstrates exceptional accuracy in the 1d Wave ($c = 2.0$) case, where the errors reach the remarkably low level of $9 \times 10^{-6}$. These results also highlight the framework's robustness across different physical regimes and its ability to adapt to increased complexity through $r$. For instance, in the high-frequency Wave $c = 5.0$, we employ a higher rank $r = 16$ to effectively capture the intricate oscillatory patterns that challenge standard architectures. A detailed sensitivity analysis of r and its impact on convergence is provided in Section 4.6. To ensure the statistical significance of these results, all reported relative $\ell_2$ errors and training times are averaged over five independent runs with different random seeds.

Table 3: Baseline comparison for Poisson equations (5d and 10d). Relative $\ell_2$ errors are reported in units of $\times 10^{-4}$.

| Problem | Model | Params | Epochs | Time (s) | $\ell_2$ Error $(10^{-4})$ |
|---|---|---|---|---|---|
| Poisson (5d) | vanilla PINN | 21249 | 21000 | 165.73 | 33.33 |
| Poisson (5d) | unshared MLP | 21765 | 10400 | 390.46 | 1.90 |
| Poisson (5d) | **shared MLP** | 4417 | 10200 | 390.75 | 2.64 |
| Poisson (10d) | vanilla PINN | 21569 | 31500 | 1309.24 | 580.31 |
| Poisson (10d) | unshared MLP | 46130 | 10900 | 1440.48 | 18.30 |
| Poisson (10d) | **shared MLP** | 4677 | 12300 | 1565.75 | 9.88 |

### 4.3 Dimension Decomposition and Interpretability

**Parameter efficiency of shared MLP.** We first demonstrate the structural advantages of the proposed shared MLP architecture within each expert. Table 2 compares the number of trainable parameters across different PDE benchmarks. We fix $r = 16$ for this parameter test. In these experiments, we employ a single expert for the Poisson and Wave equations, while for the more complex Viscous Burgers, ADR (Traveling Wave and Oscillatory Decay) and Allen–Cahn benchmarks, we utilize the MoE framework with $K = 2$, $K = 3$, $K = 4$ and $K = 2$ experts, respectively. A detailed ablation study regarding the choice of $r$ and $K$ is provided in Section 4.6.

The shared MLP design achieves a significant reduction in model complexity compared to the unshared MLPs baseline. The advantage enlarges as the input dimension $d$ grows, highlighting the scalability of our approach. Within a single expert module, the parameter count of the shared MLP remains invariant to the input dimension, whereas the unshared MLPs architecture exhibits a linear growth $\mathcal{O}(d)$. When integrated into the MoE framework, the shared MLP further reduces the parameter usage by (approximately $30\%-38\%$) sharing parameters across experts.

**Accuracy of Shared MLP.** We evaluate the proposed shared MLP within single expert module on the 5d and 10d Poisson equation and compare it with (i) an unshared MLP where each dimension is processed independently, and (ii) a vanilla PINN baseline. Crucially, we adopts mesh-free random sampling rather than regular rectangular tensor-product grids. This prevents direct the execution of SPINNs. Under this condition, our "unshared MLP" baseline serves as a natural mathematical variant of SPINNs. We use $r = 1$ and $r = 5$ for 5d and 10d case respectively. For the PINN baseline, we use a 6-layer fully connected MLP with width 64 and `Tanh` activations. Since training terminates when the gradient norm falls below $10^{-9}$, or the loss change falls below $10^{-12}$, different methods run for different numbers of steps. For fair visualization, all curves are truncated to the minimum common training length in both epochs and wall-clock time. The quantitative results corresponding to Fig 2 are summarized in Table 3.

The relative $\ell^2$ error of 5d case versus epochs (left) and versus time (right) is illustrated in Fig 2. Both shared and unshared MLPs converge substantially faster than the vanilla PINN, reaching one order of magnitude lower error within the same optimization horizon. In comparison with the unshared MLP, the shared MLP achieves comparable accuracy in both epoch-based and time-based evaluations. However, this comparable performance is obtained with significantly fewer parameters due to cross-dimension parameter sharing, see Table 3. Hence, the shared MLP maintains learning performance while offering great improved parameter efficiency, resulting in a more favorable accuracy–complexity trade-off.

We further evaluate the proposed the shared MLP on the 10d Poisson equation, where the difficulty of optimization and approximation increases due to the curse of dimensionality. In this challenging setting, we benchmark our shared MLP against both the vanilla PINNs baseline and the unshared MLP variant. Fig 2 shows a pronounced performance gap. Our shared MLP eventually achieves two orders of magnitude error reduction, while the PINN baseline exhibits slow convergence and remains at a significantly higher error level under larger number of parameters. Under standard zero-bias initialization, the unshared MLP suffers from numerical vanishing due to the multiplication of decoupled coordinate features. This collapse prevents backpropagation and completely stalls the optimization process. To ensure a rigorous and mean-ingful comparison, we set the initial biases to 1 for the unshared MLP. Even with this targeted initialization

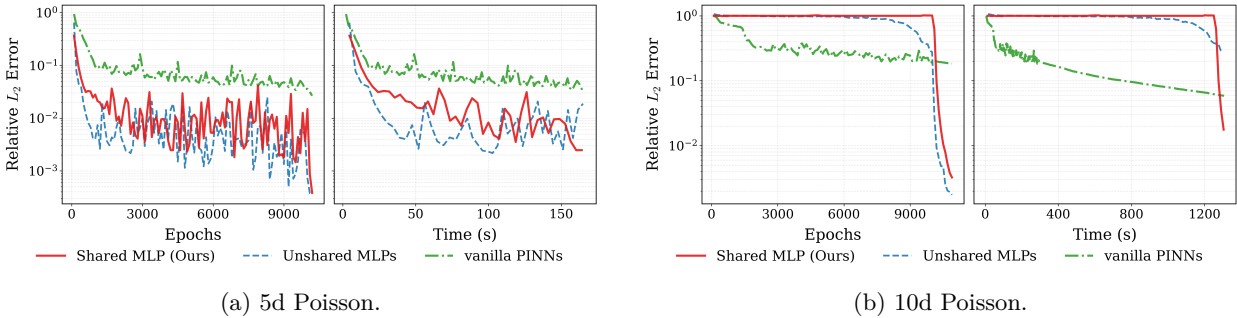

(a) 5d Poisson.                                      (b) 10d Poisson.

Figure 2: **Relative $\ell_2$ error (log scale) for Poisson equations.** Left: error versus training epochs. Right: error versus wall-clock time. Parameter sharing maintains accuracy while increasing parameter efficiency.

assistance, our shared MLP outperforms the unshared MLP baseline. Ultimately, these results underscore that dimension-wise parameter sharing is not merely a strategy for model compression, but serves as an effective inductive bias to enhances the model's capacity in high-dimensional regimes.

**Cross-Dimensional Transferability of Shared MLP.** The shared MLP architecture offers a significant advantage in transferability, allowing learned features to be reused across different dimensionalities. To evaluate this, we fix $r = 5$ pre-train a model on the 5d Poisson equation and use its parameters to initialize models for 10d and 15d problems. For comparison, we also train a 10d model from scratch; however, the 15d problem proved too challenging for training from scratch, consistently failing to converge (with relative $\ell_2$ errors near 1.0).

As summarized in Table 4, fine-tuning improves both efficiency and stability. For the 10d case, the fine-tuned model converges faster and achieves a lower relative $\ell_2$ error ($6.89 \times 10^{-4}$) compared to the model trained from scratch ($9.88 \times 10^{-4}$). Significantly, while training from scratch fails in 15d, our fine-tuning approach enables the model to reach a high-precision solution with an error of $1.145 \times 10^{-2}$. These results confirm that the shared MLP structure effectively captures universal low-dimensional

Table 4: Comparison of training from scratch vs. fine-tuning for Poisson equations.

| Model | Epochs | Time (s) | $\ell_2$ Error $(10^{-4})$ |
|---|---|---|---|
| 10d (src) | 12300 | 1565.75 | 9.88 |
| 10d (fine) | 11500 | 1481.66 | 6.89 |
| 15d (fine) | 10200 | 2891.88 | 114.83 |

patterns that serve as robust priors for higher-dimensional problems. Fig 3 further illustrates the training dynamics. These dynamics further support that the shared MLP captures transferable low-dimensional structures, which provide an effective initialization and substantially improve optimization stability in higher-dimensional settings. Furthermore, we evaluate our approach on Green's functions governed by diverse physical systems across multiple spatial dimensions, see Appendix D.2 for details.

**Visualizing Latent Component Trajectories.** To establish an intuitive foundation for the Variable Interpretability (VI) metric discussed in Section 4.3, we visualize the dimension-wise learning performance using the 1d Wave with the wave speed $c = 2.0$. Given the analytical solution $u(t, x) = \sin(\pi x)\cos(c\pi t)$, the target function is inherently separable. We therefore employ a single expert module with $r = 1$ to verify if the framework can recover the exact temporal component $f_t(t) = \cos(c\pi t)$ and spatial component $f_x(x) = \sin(\pi x)$. Fig 4 illustrates the evolution of the learned components at training steps 1000, 2000, 3000, and 4000. A distinct learning

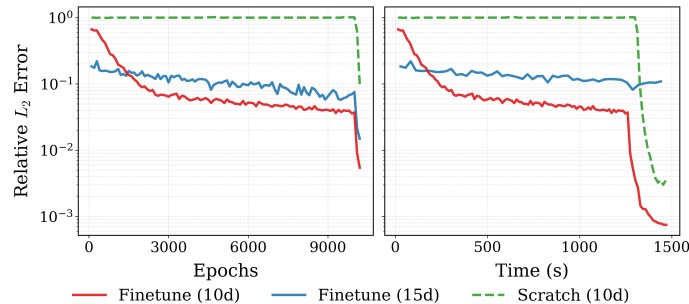

Figure 3: **Fine-tuning Comparison.** Relative $L_2$ error vs. training epochs (left) and wall-clock time (right). Fine-tuning demonstrates superior efficiency.

difference is observed: the spatial component
$f_x(x)$, possessing a lower frequency, is accurately captured within the first 1000 steps. In contrast, the temporal component $f_t(t)$, with a higher frequency, requires up to 4000 steps to converge. This phenomenon is consistent with the well-known spectral bias Rahaman et al. (2019); Xu et al. (2025) of neural networks, where high-frequency modes are intrinsically harder to approximate.

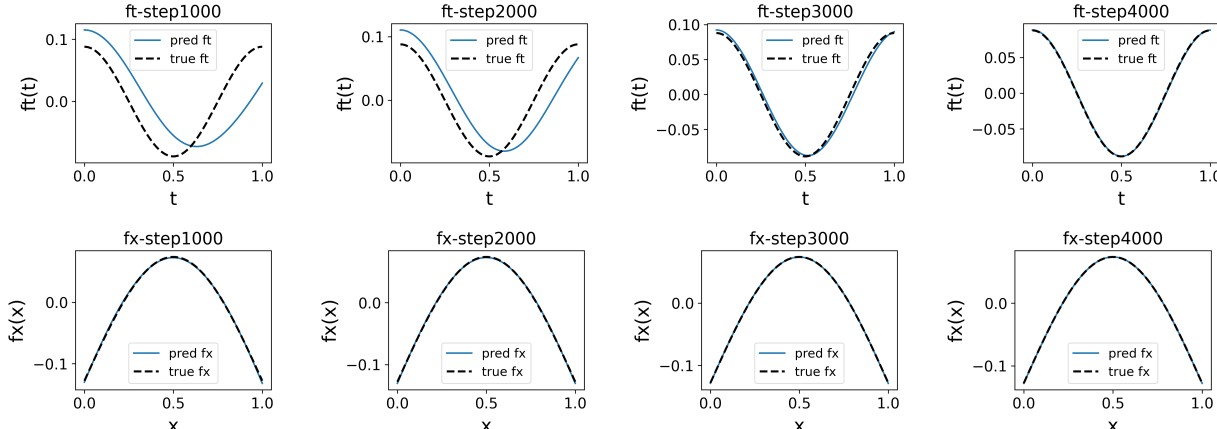

Figure 4: **Components' interpretability of 1d Wave equation when** $c = 2.0$**.** The first row represents comparison of $t$-component while the second row represents comparison of $x$-component. Here "pred $f_x$" and "pred $f_t$" in the figure refers to the shared MLP processing $x$ and $t$ respectively. The black dotted line stands for true value from analytical solution and the blue solid line stands for predicted value. From left to right, the columns represent the 1000th, 2000th, 3000th, and 4000th training steps, respectively.

**Quantitative Assessment of VI.** Table 5 summarizes the mean variable importance (VI) over dimension components for different values of $r$, averaged across five independent random seeds.

For the 5d Poisson equation with analytical solution $u = \prod_{i=1}^{5} \sin(\pi x_i)$, one might expect that $r = 1$ would already capture the separable structure. However, the empirical results indicate that $r = 1$ is insufficient for interpretability, yielding very low VI. As $r$ increases, VI improves rapidly: by $r = 3$ it already exceeds 99%, and by $r = 4$–5 it effectively reaches 100%. This trend demonstrates that a small but nontrivial rank is necessary to fully recover the intrinsic separable structure.

A similar phenomenon is observed for the 10d Poisson problem. Although the dimensionality doubles, full interpretability is still achieved with $r = 5$, where VI approaches 100% with very small variance across seeds. This indicates that the required rank grows slowly with dimension and remains modest even in higher-dimensional settings.

For the wave equation, the required $r$ depends on the solution complexity. In the 1d wave case with $c = 2.0$, the solution structure is sufficiently simple that VI remains 100% for all tested $r$. In contrast, when $c = 5.0$, the solution contains higher-frequency temporal components $\cos(c\pi t)$, increasing structural complexity. Consequently, $r = 1$ no longer suffices, but VI steadily improves as $r$ increases and approaches 100% at $r = 5$.

For the 2d wave equation with analytical solution $u(t, x_1, x_2) = \sin(\pi x_1) \sin(\pi x_2) \cos(\sqrt{2}c\pi t)$ and $c = 2.0$, the additional spatial dimension further increases representation complexity. Nevertheless, VI again approaches 100% when $r = 5$, indicating that relatively small ranks are sufficient to recover the dominant variable interactions.

Finally, for the ADR with oscillatory decay, which is modeled using a MoE architecture with two experts, the reported VI values correspond to the average importance over the two experts. The overall trend remains consistent: increasing $r$ systematically enhances interpretability.

Table 5: Mean $VI$ in percentage from ($\times 100$) over dimension components for different values of $r$ across the PDE examples. All values are averaged over 5 seeds.

| PDE examples | r=1 | r=2 | r=3 | r=4 | r=5 |
|---|---|---|---|---|---|
| 5d Poisson | $4.11 \pm 0.00$ | $91.21 \pm 12.66$ | $99.72 \pm 0.14$ | $99.99 \pm 0.01$ | $100.00 \pm 0.00$ |
| 10d Poisson | $4.82 \pm 1.10$ | $87.48 \pm 7.49$ | $99.46 \pm 0.06$ | $99.99 \pm 0.01$ | $100.00 \pm 0.00$ |
| 1d Wave $c = 2.0$ | $100.00 \pm 0.00$ | $100.00 \pm 0.00$ | $100.00 \pm 0.00$ | $100.00 \pm 0.00$ | $100.00 \pm 0.00$ |
| 1d Wave $c = 5.0$ | $49.26 \pm 1.04$ | $83.09 \pm 2.82$ | $90.65 \pm 6.78$ | $90.72 \pm 6.64$ | $99.40 \pm 0.10$ |
| 2d Wave $c = 2.0$ | $67.56 \pm 1.34$ | $94.53 \pm 3.10$ | $99.74 \pm 0.19$ | $99.97 \pm 0.02$ | $100.00 \pm 0.00$ |
| ADR (Oscillatory Decay) | $43.58 \pm 16.58$ | $75.71 \pm 9.09$ | $90.16 \pm 2.02$ | $95.12 \pm 1.11$ | $96.08 \pm 0.45$ |

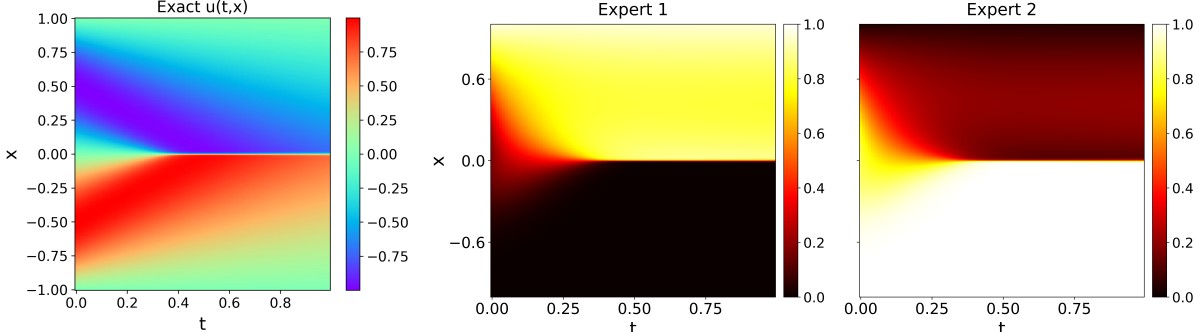

Figure 5: **Exact solution and domain decomposition results for the Viscous Burgers equation with $K = 2$ experts and viscosity $\nu = \frac{0.01}{\pi}$.** From left to right: exact solution, gating weight of Expert 1, and gating weight of Expert 2. The gating network automatically identifies the shock location at $x = 0$, where the solution exhibits a sharp transition, demonstrating that the learned decomposition aligns with the underlying physical structure.

Collectively, these results confirm that modest values of $r$ are sufficient to achieve strong interpretability across different PDE families and dimensionalities, while preserving stable behavior across random initializations.

## 4.4 Adaptive Domain Decomposition

**Viscous Burgers.** Multiple experts and a router are employed for automatic domain decomposition for Viscous Burgers. Within each expert, dimension decomposition is applied. The training data consists of 10,000 randomly sampled collocation points, 256 initial points, and 200 boundary points. For testing, we adopt high-accuracy dataset generated in MATLAB, as in PINNs Raissi et al. (2019).

For the tested viscosity $\nu = \frac{0.01}{\pi}$, the solution of the viscous Burgers develops a sharp internal layer centered at $x = 0$. Although the solution remains continuous for $\nu > 0$, the steep gradient near the shock effectively separates the domain into two regions with markedly different local regularity. This makes $x = 0$ a natural and physically meaningful candidate for the splitting interface in a domain decomposition framework.

To further validate the optimization efficiency and accuracy of 3D framework, we conduct a comparative study against state-of-the-art physics-informed baseline methods. Specifically, we benchmark our model against standard PINNs (Raissi et al., 2019), APINNs (Hu et al., 2023), XPINNs (Jagtap et al., 2020c) and PirateNets (Wang et al., 2024).

The empirical convergence trajectories across training epochs are illustrated in Figure 6. As observed, 3D achieves the lowest relative $\ell_2$ error while requiring fewer parameters. In contrast, traditional XPINNs yield the highest final error. Instead, they require manually predefined subdomains and explicitly hardcoded interface conditions to enforce physical continuity across boundaries. While APINNs utilize soft-gating profiles and achieves comparable performances, they require more parameters. PirateNets leverage adaptive residual structures for robust and stable convergence. To provide a clear quantitative summary, we record the corresponding values in Table 6.

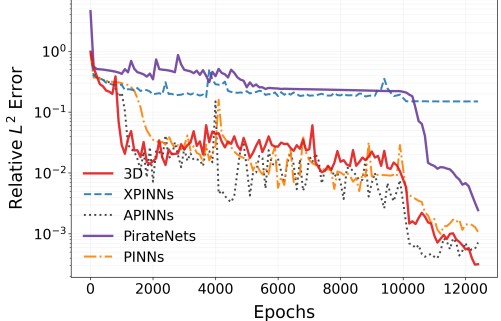

Figure 6: Convergence dynamics across different methods on the Viscous Burgers.

Table 6: Baseline comparison for Viscous Burgers. Relative $\ell_2$ errors are reported in units of $\times 10^{-4}$.

| Method | Parameters | Epochs | $\ell_2$ Error |
|---|---|---|---|
| PINN | 29377 | 12600 | 8.71 |
| APINNs | 38276 | 12400 | 6.69 |
| PirateNets | 33553 | 13900 | 9.84 |
| XPINNs | 25482 | 14500 | 1495.90 |
| 3D | 23586 | 13400 | **2.66** |

Figure 5 presents the exact solution and the learned domain decomposition results for the Viscous Burgers equation with $K = 2$ experts. The predicted solution closely matches the exact solution, and the error remains localized primarily in the vicinity of the shock, where the gradient is largest. More importantly, the gating weights exhibit a clear spatial partition aligned with the shock location: the two experts specialize on the above and below subdomains separated by $x = 0$. This behavior indicates that the model automatically identifies the principal transition and organizes the experts accordingly, demonstrating that the learned decomposition is not arbitrary but strongly correlated with the underlying solution structure. To further show the robustness of our domain decomposition mechanism, we include experimental results with noisy boundary and initial conditions in Appendix D.

**Advection Diffusion Reaction (ADR).** To further evaluate the effectiveness of the proposed automatic domain decomposition strategy, we analyze the learned gating behaviors for both Advection Diffusion Reaction (ADR) problems with traveling wave and oscillatory decay. As illustrated in Fig 7, the first column displays the exact solutions, characterized by distinct high-gradient interfaces: a diagonal wave front in the traveling wave case (left) and a horizontal transition regime at $x \approx 0.35$ in the oscillatory decay case (right). For both problems, the gating network autonomously identifies and partitions the domain along the intrinsic physical boundaries of the solution. By automatically aligning sub-domains with the solution's sharp transitions, the gating network simplifies the learning task for each individual expert. Instead of approximating a globally complex function, each expert specializes in a relatively smooth local manifold, thereby significantly enhancing the overall approximation accuracy and robustness of the ADR solver.

**Allen-Cahn.** To further evaluate the effectiveness of our model beyond problems with closed-form solutions, we consider the Allen-Cahn equation, whose analytical solution is unavailable. We use Fourier pseudo-spectral method for solving reference solution, see Appendix B.5 The equation exhibits nonlinear reaction dynamics and interface evolution, making it substantially more challenging than separable elliptic benchmarks. We impose the initial condition through a hard constraint following Hao et al. (2025). Although the solution is non-separable and therefore VI cannot be computed in this case, our model remains highly accurate. With only $K = 2$ experts, the method achieves a relative $\ell_2$ error of $6 \times 10^{-3}$, demonstrating strong approximation capability even in the absence of analytical structure. This experiment highlights the robustness and general applicability of our framework.

In addition, automatic domain decomposition emerges clearly in this example (see Fig 8). The two experts specialize in distinct spatial regions separated by the evolving interface. One expert predominantly captures the central transition layer, while the other focuses on the outer bulk phases. The learned partition dynamically adapts along the space–time domain, aligning with the intrinsic phase-separation structure of the solution. This behavior indicates that the gating mechanism successfully identifies physically meaningful subdomains without any prior domain partition, providing empirical evidence that our model performs adaptive domain decomposition.

## 4.5 Synergy Verification

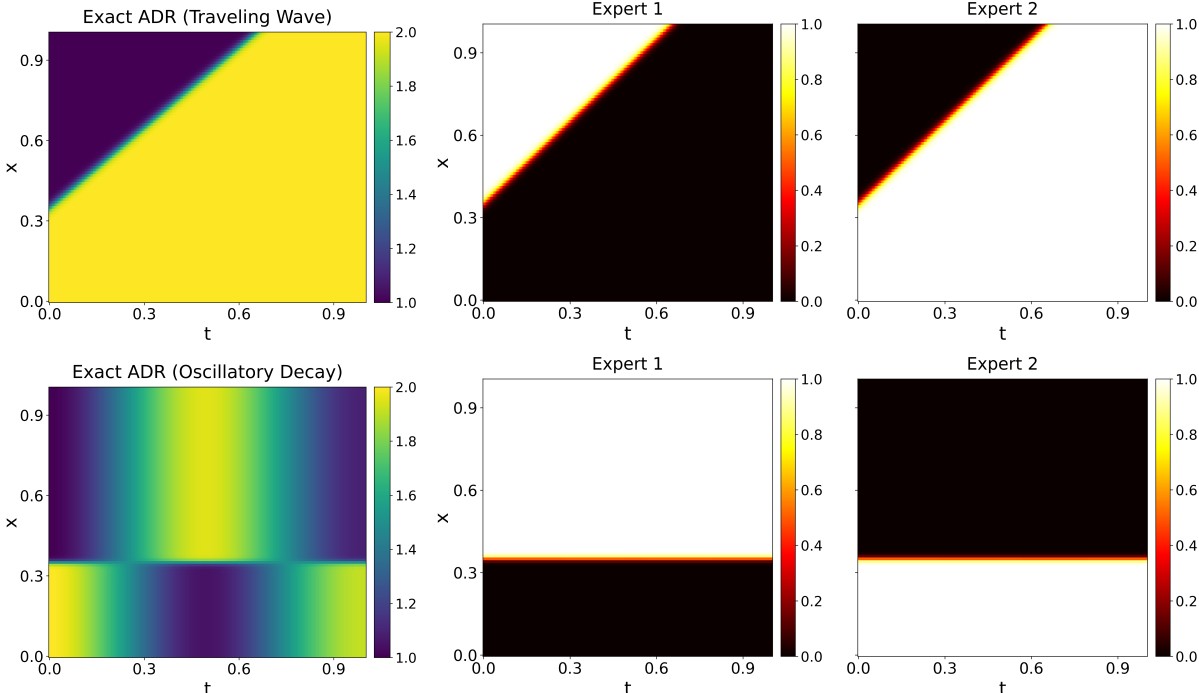

Figure 7: **Exact solution and domain decomposition results for ADR problems with $K = 2$.** First row: ADR with traveling wave. Second row: ADR with oscillatory decay. From left to right: exact solution, gating weight of Expert 1, and gating weight of Expert 2. In both cases, the learned decomposition interfaces align closely with the sharp physical transitions in the solution, demonstrating that the gating network captures the underlying dynamics.

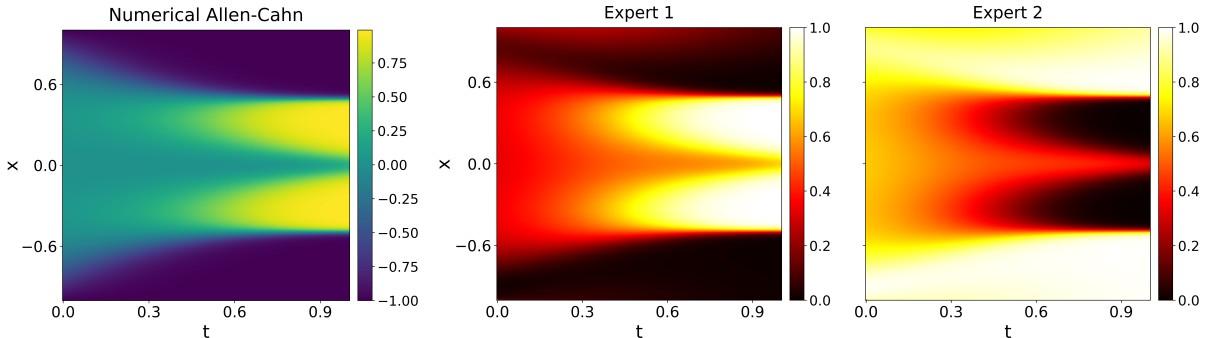

Figure 8: **Numerical solution and learned domain decomposition for the Allen–Cahn equation with $K = 2$ experts.** From left to right: numerical solution, gating weight of Expert 1, and gating weight of Expert 2. The learned gating weights reveal an adaptive partition that aligns with the interface dynamics, demonstrating automatic domain decomposition without predefined subdomains.

To thoroughly substantiate our claim that the unified 3D framework yields a synergistic enhancement rather than a simple superposition of two methods, we design a challenging benchmark that demands both decomposition capabilities. We consider a 5d nonlinear Poisson equation featuring an oblique, sharp internal transition front that is misaligned with the standard Cartesian coordinate axes. PDE details are illustrated in Appendix B.

To evaluate the synergy, we contrast our 3D framework against two isolated ablations:

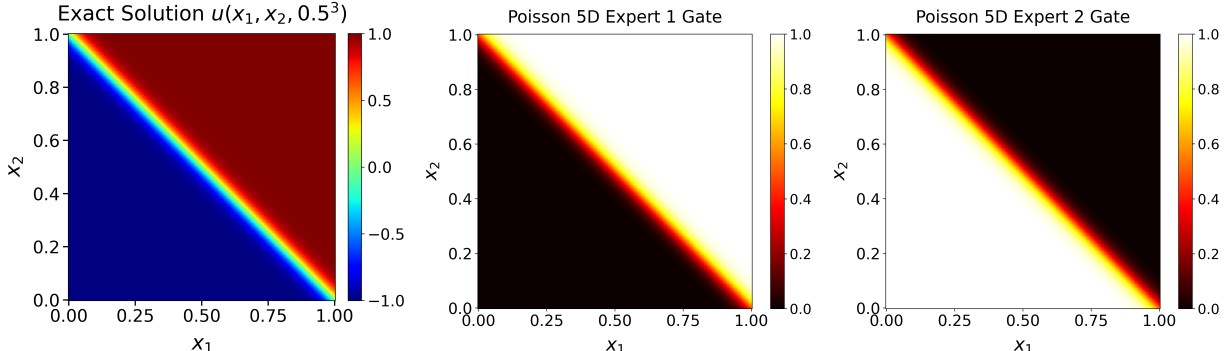

Figure 9: Numerical solution and learned domain decomposition for the synergic 5d Poisson with $K = 2$.

1. **Dimension Decomposition Only:** Disables the MoE gating mechanism, forcing a single shared MLP to capture the highly localized shock.
2. **Domain Decomposition Only:** Disables the dimension-wise shared-MLP structure, utilizing standard unshared multi-variable MLPs inside each expert.

Our empirical findings, shown in Table 7, show that isolated strategies fail to converge satisfactorily. Under identical configurations with rank $r = 5$, the *Dimension Decomposition Only* strategy exhibits severe underfitting, as a globally decoupled coordinate framework lacks the localized capacity to reconstruct sharp oblique discontinuities. On the other hand, the *Domain Decomposition Only* approach exhibits poorer performance with larger model capacity. In contrast, our unified **3D framework** achieves the best relative $\ell_2$ error. By utilizing the shared MLP structure, the dimension decomposition compresses the input parameter space to bypass the curse of dimensionality, while the adaptive MoE gating network organically breaks down the non-convex shock region into distinct functional subdomains, as shown in Figure 9. This compellingly confirms that our dimension-wise and domain-wise co-decomposition mechanisms work in strong synergy to solve complex problems that are otherwise intractable by either method alone.

### 4.6 Ablation and Sensitivity Studies

### 4.6.1 Rank Sensitivity Analysis on $r$

In this section, we investigate the influence of the hyperparameter $r$ on the model's approximation capability and convergence behavior. The $r$ in our model, see equation 3.3, directly impact the expressive power of the functional decomposition. We evaluate the performance across all PDE benchmarks by setting $r \in \{1, 2, 4, 5, 8, 10, 16, 32\}$ according to different PDE problems. The results including epoch

Table 7: Ablation study on the components of 3D.

| Methods | Parameters | Epochs | Error |
|---|---|---|---|
| 3D | 22348 | 11900 | $7.69 \times 10^{-4}$ |
| Domain Only | 23076 | 13000 | $1.37 \times 10^{-3}$ |
| Dimension Only | 4677 | 17300 | $5.44 \times 10^{-1}$ |

plots and time plots, illustrated in Fig 10, reveal how the solution's structural complexity dictates the required $r$. The detailed results are presented in Table 8.

**Efficiency on Simple Separable Solution Manifolds.** For problems whose solutions lie on simple low-rank manifolds, the model achieves high rank $r$ efficiency.

**Poisson (5d and 10d):** The Poisson problem is inherently separable (Section 4.3). Despite slower convergence and slightly higher errors for the 10d case, even a minimal rank ($r = 1$) performs well. For 5d, all tested ranks converge to an error floor of $\approx 10^{-4}$, while for 10d, the floor is $\approx 10^{-3}$, demonstrating robustness across dimensions.

**Wave ($c = 2.0$):** Similarly separable (Section 4.3), the low-frequency wave ($c = 2.0$) is captured perfectly with $r = 1$, achieving a relative $\ell_2$ error of $6 \times 10^{-6}$. As shown in Table 5 and Fig 4, $r = 1$ leads to $VI = 100\%$,

Table 8: Rank ablation across PDE benchmarks. For each PDE, multiple ranks $r$ are evaluated. We report the training epochs, total time in seconds, and the relative $\ell_2$ error ($\times 10^{-4}$).

| PDE | Rank $r$ | Epochs / Time (s) | $\ell_2$ error ($\times 10^{-4}$) |
|---|---|---|---|
| 5d Poisson | 1 | 10100 / 381.72 | 3.62 |
| | 5 | 10600 / 393.26 | 3.71 |
| | 10 | 11600 / 449.01 | **1.17** |
| | 16 | 10700 / 402.89 | 3.91 |
| 10d Poisson | 1 | 11400 / 1444.05 | 15.31 |
| | 5 | 12300 / 1565.75 | **9.88** |
| | 10 | 10700 / 1481.40 | 25.25 |
| | 16 | 11100 / 1637.08 | 10.43 |
| Wave ($c = 2.0$) | 1 | 10500 / 138.59 | **0.06** |
| | 5 | 12000 / 167.72 | 1.93 |
| | 10 | 11000 / 150.82 | 6.05 |
| | 16 | 12900 / 178.35 | 3.34 |
| Wave ($c = 5.0$) | 1 | 10000 / 130.15 | 11242.26 |
| | 5 | 16200 / 246.45 | 5684.20 |
| | 10 | 25500 / 424.80 | 210.65 |
| | 16 | 20800 / 343.23 | **61.42** |
| Burgers | 4 | 19600 / 555.13 | 344.49 |
| | 8 | 14900 / 382.39 | 3.67 |
| | 16 | 14300 / 360.88 | **1.34** |
| | 32 | 15900 / 416.87 | 5.60 |
| ADR (Traveling) | 1 | 10200 / 331.33 | 2241.76 |
| | 2 | 11500 / 372.19 | **7.05** |
| | 4 | 11700 / 380.11 | 8.71 |
| | 16 | 10600 / 342.56 | 10.54 |
| ADR (Oscillatory) | 1 | 17600 /776.37 | 4.18 |
| | 2 | 14100 / 658.24 | 3.91 |
| | 4 | 11100 / 496.39 | 5.84 |
| | 16 | 15800 / 686.08 | **2.67** |
| Allen Cahn | 1 | 23200 / 1019.94 | 312.84 |
| | 4 | 24700 / 1017.79 | 102.08 |
| | 16 | 20200 / 884.33 | **65.86** |
| | 32 | 24800 / 1090.48 | 82.22 |

indicating each dimension component is learned accurately. Higher ranks provide no improvement, making $r = 1$ the optimal choice.

**ADR (Traveling Wave):** A clear transition occurs between $r = 1$ and $r \geq 2$. With $r = 1$, the model lacks sufficient expressivity, resulting in poor accuracy. Increasing to $r \geq 2$ stabilizes learning, and further rank increments have minimal impact. Higher ranks introduce oscillatory spikes in the training error, reflecting increased optimization variability.

**ADR (Oscillatory Decay):** All tested ranks achieve similar performance. The model exploits the decaying temporal modulation to effectively separate spatial and temporal features, making additional rank unnecessary.

**Capacity Requirements for Complex Solutions.** For PDEs with challenging conditions (high-frequency components) or non-separable structures, model performance is highly sensitive to the rank $r$.

**Wave ($c = 5.0$):** Increasing the frequency introduces significant difficulty. Low ranks ($r = 1$ or $5$) fail to learn, stagnating at high relative error. A notable improvement occurs at $r = 10$, reducing the error to

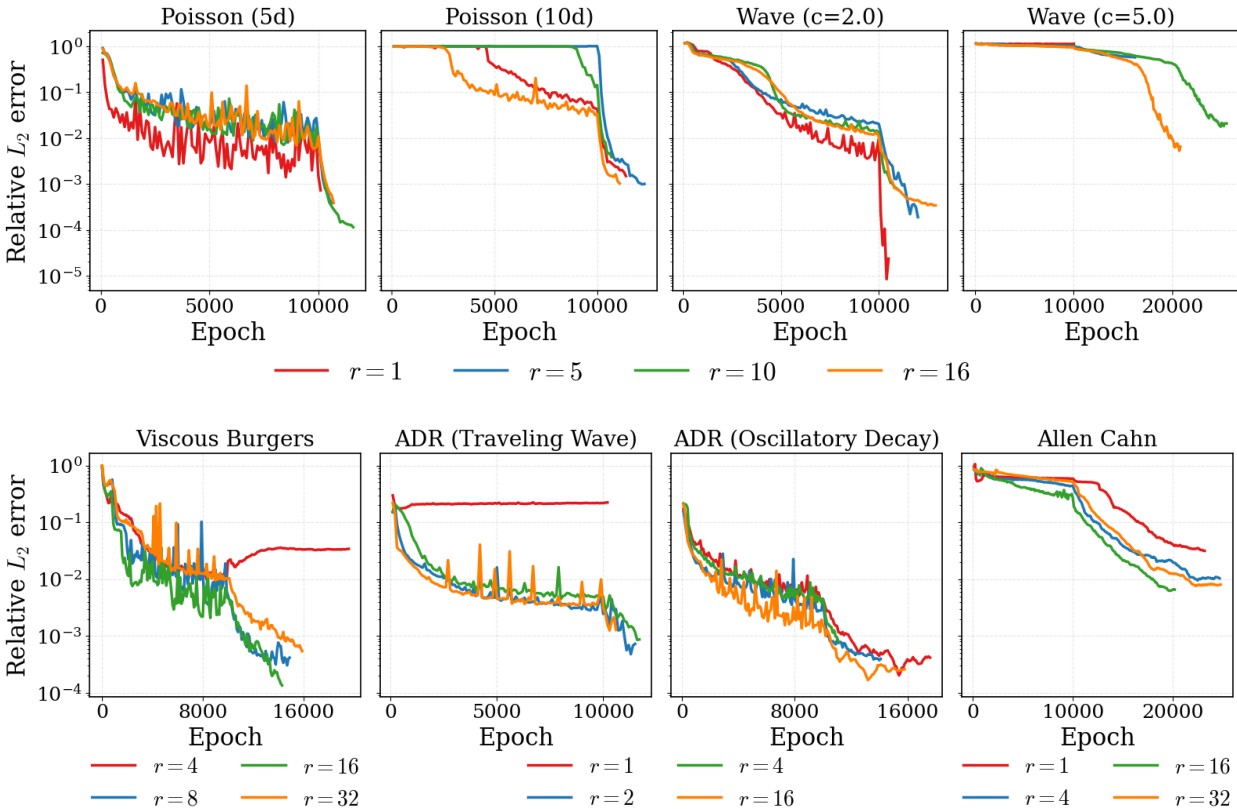

Figure 10: **Performance evaluation (relative $L_2$ error vs. training epochs) across PDE benchmarks under varying rank $r$.**

$2.11 \times 10^{-2}$, while $r = 16$ achieves the best precision of $6.14 \times 10^{-3}$. This illustrates that the error floor is strongly constrained by $r$ for high-frequency solution manifolds.

**Viscous Burgers:** Shock waves challenge low-rank approximations. As shown in Fig 10, $r = 4$ is insufficient to resolve high-gradient regions, leading to poor convergence. Increasing to $r = 16$ achieves $1.34 \times 10^{-4}$, while $r = 32$ provides diminishing returns and introduces optimization noise, indicating $r = 16$ as the optimal rank between training performance and efficiency.

**Allen-Cahn:** Lacking a closed-form solution and involving complex phase-field dynamics, this PDE tests the framework's generalization. Fig 10 shows $r = 1$ underfits the interface evolution ($3.13 \times 10^{-2}$ error), $r = 4$ improves approximation ($1.02 \times 10^{-2}$), and $r = 16$ yields the best result ($6.59 \times 10^{-3}$), demonstrating effective manifold learning with modest rank.

Overall, the optimal $r$ reflects the intrinsic separation rank of the PDE solution. Simple separable problems are well captured with $r \in [1, 5]$, balancing accuracy and efficiency. For non-separable or complex problems, moderate ranks ($r \approx 16$) provide sufficient expressivity, highlighting the framework's representational efficiency across varying complexity levels.

### 4.6.2 Expert Number Sensitivity Analysis on $K$

In this section, we investigate the sensitivity of the domain decomposition behavior with respect to the number of experts $K$. All experimental settings remain identical to those in Sec. 4.4, except for the value of $K$.

Fig 11 presents the decomposition result for the Viscous Burgers equation with $K = 3$. Compared to the case $K = 2$, introducing an additional expert does not lead to effective new partition structures. The learned

Table 9: Ablation study on the number of experts $K$. We report the averaged training time and $\ell_2$ error ($\times 10^{-4}$) over 5 seeds.

| PDE | $K$ | Time (s) | $\ell_2$ Error $\times 10^{-4}$ |
|---|---|---|---|
| Viscous Burgers | 1 | 379.81 | $440.73 \pm 494.80$ |
| | 2 | 399.67 | $5.86 \pm 2.20$ |
| | 3 | 492.14 | $4.94 \pm 1.27$ |
| ADR (Traveling Wave) | 1 | 311.87 | $2465.60 \pm 8.46$ |
| | 2 | 366.79 | $17.13 \pm 10.45$ |
| | 3 | 530.01 | $16.03 \pm 10.78$ |
| ADR (Oscillatory Decay) | 1 | 376.55 | $6.70 \pm 2.16$ |
| | 2 | 430.86 | $4.99 \pm 2.44$ |
| | 3 | 563.34 | $3.87 \pm 2.14$ |
| | 4 | 737.30 | $2.51 \pm 0.80$ |
| Allen–Cahn | 1 | 856.84 | $1144.42 \pm 52.57$ |
| | 2 | 1173.62 | $155.05 \pm 123.38$ |
| | 3 | 1357.69 | $86.54 \pm 15.25$ |

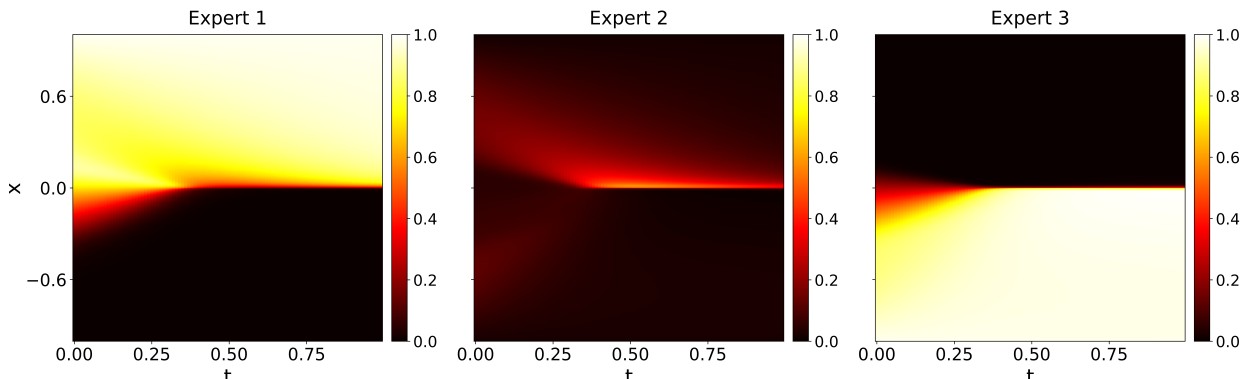

Figure 11: **Learned domain decomposition for the viscous Burgers equation with $K = 3$.** Although an additional expert is introduced, the learned partition remains aligned with the primary shock at $x = 0$, and no qualitatively new decomposition structure is observed.

decomposition remains largely consistent, indicating that two experts are already sufficient to capture the dominant structural features of the solution.

Fig 12 shows the decomposition results for the ADR equation under two settings: traveling wave (left) and oscillatory decay (right), both with $K = 3$. In both cases, the major interfaces identified when $K = 2$ persist, and the overall partition structure remains stable. This suggests that the principal dynamical regimes of the solution are already captured with a smaller number of experts. For the oscillatory decay case (right), the model appears to attempt a further subdivision of the lower spatial region ($0 < x < 0.35$) when $K$ increases. However, as seen from the exact solution (Fig 7), this region is characterized by smooth and gradually varying oscillatory behavior without sharp transitions or regime changes. Due to this intrinsic smoothness, the additional expert does not produce a clearly separated subdomain, and the resulting partition remains diffuse rather than forming a distinct interface.

To further examine this phenomenon, we increase the number of experts to $K = 4$, and the corresponding results are shown in Fig 14. Even with a larger $K$, no clear additional interface emerges in the lower region, reinforcing the observation that the decomposition is fundamentally limited by the regularity of the underlying solution rather than by model capacity.

Fig 13 shows the decomposition results for the Allen-Cahn equation with $K = 3$. Similar to the previous cases, introducing an additional expert does not yield a meaningful refinement of the partition structure.

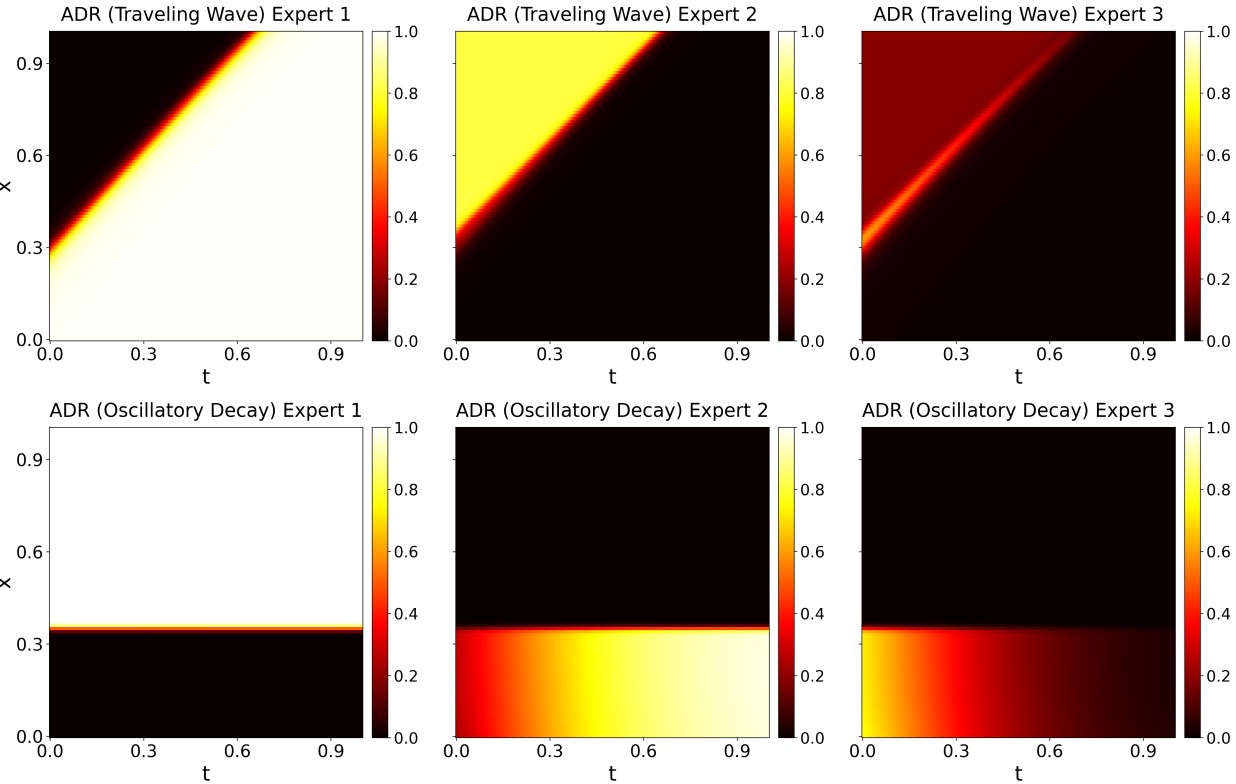

Figure 12: **Domain decomposition results for the ADR equation with $K = 3$.** First row: traveling wave case. Second row: oscillatory decay case. For both dynamical regimes, the dominant interfaces observed in the $K = 2$ setting remain stable when increasing $K$ to 3.

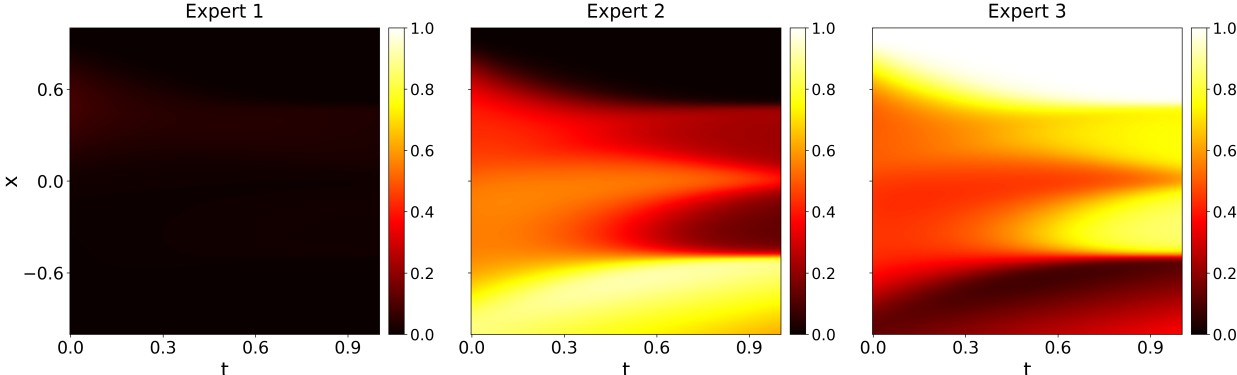

Figure 13: **Domain decomposition results for the Allen-Cahn equation with $K = 3$.** The dominant interfaces observed in the $K = 2$ setting remain stable when increasing $K$ to 3.

The dominant interface region is already well captured when $K = 2$, and the third expert receives negligible gating weight across the domain, effectively becoming inactive.

As shown in Table 9, increasing the number of experts from $K = 1$ to $K \geq 2$ leads to a reduction in $\ell_2$ error across all benchmarks, confirming the necessity of multi-expert decomposition over a single shared model. However, further increasing $K$ beyond 2 yields only marginal improvements while incurring additional computational cost. Combined with the observed domain decomposition patterns, these results suggest that

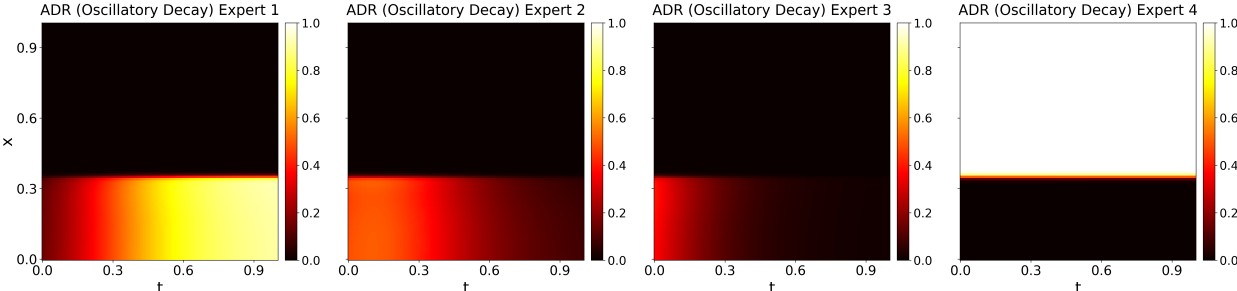

Figure 14: **Learned domain decomposition for the ADR equation with oscillatory decay under** $K = 4$**.** Despite increasing the number of experts, the decomposition remains structured around the dominant solution features, without forming new subdomains.

$K = 2$ provides the best trade-off between accuracy and efficiency, and is therefore the optimal choice in our setting.

## 5    Conclusion

In this paper, we propose Dimension Domain Co-Decomposition (3D), a PINNs-based framework that unifies dimension decomposition and MoE-driven domain decomposition. Within each expert, a shared MLP processes coordinate–index pairs to produce dimension-wise functions. To quantify the alignment between predicted dimension component and ground truth component, we introduce Variable Interpretability (VI), an dimension-wise orthogonal evaluation conditioned on the well-learned global solution. At the MoE level, the router adaptively partitions the domain so that experts specialize in local regions without requiring predefined subdomains or explicit interface conditions. Through experiments on PDE benchmarks, we show that 3D not only achieves good accuracy but also produces interpretable decompositions across dimensions according to VI. Nevertheless, our study has limitations. VI relies on reference solutions that are dimension-separable. For non-separable solutions, one possible extension would be to first construct approximate reference factors directly from the solution itself, for example via low-rank or spectral surrogate factorizations, and then evaluate the alignment between the learned and reference subspaces. However, such an approach inevitably introduces additional approximation error and ambiguity regarding the choice of factorization, so the resulting VI would no longer represent a clean ground-truth alignment measure. We view developing a principled extension of VI to general non-separable solutions as a valuable direction for future work. Furthermore, VI can be extended in a continuous manner in $L^2$ spaces. Our current discrete formulation can be viewed as a numerical approximation of its continuous counterpart. A mathematically rigorous continuous treatment, drawing upon the generalized frame and numerical approximation theory(Adcock & Huybrechs, 2019), remains an open trajectory for future work.

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

# A   Proof of Theorem 1

**Lemma 1.** *Let $\Omega = \prod_{j=1}^{d} \Omega_j \subset \mathbb{R}^d$ be a compact domain, where each $\Omega_j \subset \mathbb{R}$ is compact. Define the algebraic tensor product space of continuous functions by*

$$\mathcal{A} = \left\{ v(\mathbf{x}) = \sum_{i=1}^{r} \prod_{j=1}^{d} g_{i,j}(x_j) \ \middle| \ g_{i,j} \in C(\Omega_j), \ r \in \mathbb{N}^{+} \right\}.$$

*Then $\mathcal{A}$ is dense in $L^2(\Omega)$.*

*Proof.* We first show that the set $\mathcal{A}$ can approximate any continuous function on $\Omega$. To this end, we verify the three conditions of the Stone–Weierstrass theorem for real-valued functions.

1. **$\mathcal{A}$ is an algebra.** Let $v_1, v_2 \in \mathcal{A}$. Then each can be written as a finite sum of separable functions:

$$v_1(\mathbf{x}) = \sum_{i=1}^{r_1} \prod_{j=1}^{d} g_{i,j}(x_j), \quad v_2(\mathbf{x}) = \sum_{k=1}^{r_2} \prod_{j=1}^{d} h_{k,j}(x_j).$$

   Their sum and product can be expanded into finite sums of products of continuous functions. Moreover, scalar multiplication preserves this structure. Hence, $\mathcal{A}$ is closed under addition, multiplication, and scalar multiplication, and is therefore an algebra.

2. **$\mathcal{A}$ is non-vanishing.** By choosing $g_{i,j}(x_j) \equiv 1$ for all $j$, we obtain the constant function $1 \in \mathcal{A}$. Hence, $\mathcal{A}$ contains constants.

3. **$\mathcal{A}$ separates points.** Let $\mathbf{x}, \mathbf{y} \in \Omega$ with $\mathbf{x} \neq \mathbf{y}$. Then there exists at least one coordinate $k$ such that $x_k \neq y_k$. Since $\Omega_k$ is compact, we can choose a function $g_k \in C(\Omega_k)$ such that $g_k(x_k) \neq g_k(y_k)$ (for example, $g_k(z) = z$). For all $j \neq k$, set $g_j \equiv 1$. Then the function

$$v(\mathbf{z}) = g_k(z_k) \prod_{j \neq k} 1 \in \mathcal{A}$$

   satisfies $v(\mathbf{x}) \neq v(\mathbf{y})$. Thus, $\mathcal{A}$ separates points.

Then by Stone-Weierstrass theorem, for any $w \in C(\Omega)$ and any $\delta > 0$, there exists a $v \in \mathcal{A}$ such that:

$$\|w - v\|_{\infty} = \sup_{\mathbf{x} \in \Omega} |w(\mathbf{x}) - v(\mathbf{x})| < \delta$$

Because $\Omega$ is a compact subset of $\mathbb{R}^d$, it has a finite Lebesgue measure $\mu(\Omega) < \infty$. The $L^2$ norm is bounded by the $L^\infty$ norm:

$$\|w - v\|_{L^2(\Omega)} = \left( \int_\Omega |w(\mathbf{x}) - v(\mathbf{x})|^2 d\mathbf{x} \right)^{1/2} \le \left( \int_\Omega \delta^2 d\mathbf{x} \right)^{1/2} = \delta \sqrt{\mu(\Omega)}$$

Thus, $\mathcal{A}$ is also dense in $C(\Omega)$ with respect to the $L^2$ metric.

The space of continuous functions with compact support, $C_c(\Omega)$, is dense in $L^p(\Omega)$ for $1 \le p < \infty$. Since $\Omega$ itself is compact, $C_c(\Omega) = C(\Omega)$. Therefore, for any target function $u \in L^2(\Omega)$ and any $\varepsilon > 0$, there exists a continuous function $w \in C(\Omega)$ such that:

$$\|u - w\|_{L^2(\Omega)} < \frac{\varepsilon}{2}$$

Let $u \in L^2(\Omega)$ and $\varepsilon > 0$. Choose $w \in C(\Omega)$ such that $\|u - w\|_{L^2(\Omega)} < \frac{\varepsilon}{2}$. Then choose $v \in \mathcal{A}$ such that $\|w - v\|_\infty < \frac{\varepsilon}{2\sqrt{\mu(\Omega)}}$. Consequently, $\|w - v\|_{L^2(\Omega)} < \frac{\varepsilon}{2}$.

By the triangle inequality:

$$\|u - v\|_{L^2(\Omega)} \le \|u - w\|_{L^2(\Omega)} + \|w - v\|_{L^2(\Omega)} < \frac{\varepsilon}{2} + \frac{\varepsilon}{2} = \varepsilon$$

$\square$

Now we prove the Theorem 1 in the main paper.

**Theorem** ( Universal Approximation with Shared MLP). *Let $\Omega \subset \mathbb{R}^d$ be compact and let $u \in L^2(\Omega)$. Then for any $\varepsilon > 0$, there exists an integer $r > 0$ and a neural network $f : \mathbb{R} \times \{0, 1, \ldots, d-1\} \to \mathbb{R}^r$ such that*

$$\hat{u}(\mathbf{x}) = \frac{1}{r} \sum_{i=1}^{r} \prod_{j=1}^{d} f^{(i)}(x_j, j-1)$$

*satisfies*

$$\|u - \hat{u}\|_{L^2(\Omega)} < \varepsilon.$$

*Proof.* Given $u \in L^2(\Omega)$ and $\varepsilon > 0$, by Lemma 1, there exists a finite rank $r$ and continuous functions $g_{i,j}(x_j)$ such that:

$$\left\| u(\mathbf{x}) - \sum_{i=1}^{r} \prod_{j=1}^{d} g_{i,j}(x_j) \right\|_{L^2(\Omega)} < \frac{\varepsilon}{2}$$

Since each $\Omega_j$ is compact and $g_{i,j}$ is continuous, there exists a uniform bound $M$ such that $|g_{i,j}(x_j)| \le M$ for all $i, j$.

Let $X = \bigcup_{j=1}^{d} \Omega_j$ (a compact subset of $\mathbb{R}$ ). Define the input domain for our network as $K = X \times \{0, 1, \ldots, d-1\} \subset \mathbb{R}^2$. We define the target function $G : K \to \mathbb{R}^r$ such that the $i$-th component is:

$$G^{(i)}(x, k) = r^{\frac{1}{d}} \cdot g_{i,k+1}(x)$$

This construction ensures that $\frac{1}{r} \prod_{j=1}^{d} G^{(i)}(x_j, j-1) = \prod_{j=1}^{d} g_{i,j}(x_j)$ for any $x_j \in \Omega_j$.

The domain $K$ is a union of finitely many compact line segments in $\mathbb{R}^2$, thus $K$ is compact. By the universal approximation theorem Cybenko (1989), for any $\delta > 0$, there exists a neural network $f : K \to \mathbb{R}^r$ such that:

$$\|f - G\|_\infty < \delta \implies \max_{i,j} \left| f^{(i)}(x_j, j-1) - r^{\frac{1}{d}} g_{i,j}(x_j) \right| < \delta$$

Let $\tilde{g}_{i,j} = r^{\frac{1}{d}} g_{i,j}$. Using the property that for bounded values $|a_j|, |b_j| \leq \mathcal{M}$, the product error is bounded by $|\prod a_j - \prod b_j| \leq d\mathcal{M}^{d-1} \max |a_j - b_j|$. Suppose $|\tilde{g}_{i,j}| \leq \tilde{M}$, then we obtain:

$$\left| \prod_{j=1}^{d} f^{(i)}(x_j, j-1) - \prod_{j=1}^{d} \tilde{g}_{i,j}(x_j) \right| \leq d(\tilde{M} + \delta)^{d-1}\delta$$

By choosing $\delta$ sufficiently small, we can ensure:

$$\left| \prod_{j=1}^{d} f^{(i)}(x_j, j-1) - \prod_{j=1}^{d} \tilde{g}_{i,j}(x_j) \right| < \frac{r\varepsilon}{2\sqrt{\mu(\Omega)}}$$

Summing over $i$ and multiplying by $\frac{1}{r}$ :

$$\left| \hat{u}(\mathbf{x}) - \sum_{i=1}^{r} \prod_{j=1}^{d} g_{i,j}(x_j) \right| < \frac{\varepsilon}{2\sqrt{\mu(\Omega)}}$$

Integrating the square of the error over $\Omega$ gives the $L^2$ distance:

$$\left\| \hat{u} - \sum_{i=1}^{r} \prod_{j=1}^{d} g_{i,j}(x_j) \right\|_{L^2(\Omega)} < \frac{\varepsilon}{2}$$

Finally, by the triangle inequality:

$$\|u - \hat{u}\|_{L^2(\Omega)} \leq \left\| u - \sum_{i=1}^{r} \prod_{j=1}^{d} g_{i,j}(x_j) \right\|_{L^2} + \left\| \sum_{i=1}^{r} \prod_{j=1}^{d} g_{i,j}(x_j) - \hat{u} \right\|_{L^2} < \frac{\varepsilon}{2} + \frac{\varepsilon}{2} = \varepsilon$$

$\square$

## B  Details of PDE Examples

In this appendix, we detail the PDE setups used in the main paper: Poisson, Wave, Viscous Burgers, Advection-Diffusion-Reaction and Allen-Cahn.

### B.1  Poisson

#### B.1.1  Poisson for Dimension Decomposition

We consider the Poisson problem with homogeneous Dirichlet boundary conditions:

$$\begin{cases} -\Delta u(\mathbf{x}) = f(\mathbf{x}), & \mathbf{x} \in \Omega, \\ u(\mathbf{x}) = 0, & \mathbf{x} \in \partial\Omega. \end{cases} \tag{B.1}$$

where $\Omega = [0,1]^d$ and $\mathbf{x} = (x_1, \ldots, x_d)$. We use the manufactured solution

$$u(\boldsymbol{x}) = \prod_{i=1}^{d} \sin(\pi x_i), \tag{B.2}$$

for which

$$-\Delta u = d\pi^2 \prod_{i=1}^{d} \sin(\pi x_i) = f(\boldsymbol{x}). \tag{B.3}$$

In this paper, we mainly test 5d Poisson and 10d Poisson.

### B.1.2  Poisson for Synergy Verification

To show the necessity of combining dimension decomposition with domain decomposition, we tested on the following 5d Poisson case:

$$-\Delta u(\mathbf{x}) = f(\mathbf{x}), \quad \mathbf{x} \in (0,1)^d, \tag{B.4}$$

subject to Dirichlet boundary conditions. The exact solution is analytically constructed as:

$$u_{\text{true}}(\mathbf{x}) = \tanh\left(\frac{\sum_{i=1}^{d} x_i - \frac{d}{2}}{\epsilon}\right) \cdot \prod_{i=1}^{d} \sin(\pi x_i), \tag{B.5}$$

where $\epsilon = 0.05$ governs the extreme steepness of the diagonal interface. The high dimensionality causes the curse of dimensionality, while the sharp shock front breaks any simple independent coordinate rendering, necessitating global domain routing.

### B.2  Wave

Wave equation is a time-dependent PDE that takes the form:

$$\begin{cases} u_{tt}(t, \mathbf{x}) = c^2 \Delta u, \quad \mathbf{x} \in (0,1)^d, \ t \in [0,1], \\ u(t, \mathbf{x}) = 0, \quad \mathbf{x} \in \partial(0,1)^d, \ t \in [0,1], \\ u(\mathbf{x}, 0) = \prod_{i=1}^{d} \sin(\pi x_i), \ u_t(\mathbf{x}, 0) = 0, \quad \mathbf{x} \in [0,1]^d. \end{cases} \tag{B.6}$$

where $c$ is the wave speed. In our experiments, we test 1d with $c = 2.0, 5.0$ and 2d with $c = 2.0$. The analytical form of Wave equation is $u(t, \mathbf{x}) = \prod_{i=1}^{d} \sin(\pi x_i) \cos(\sqrt{d}\pi c t)$.

From the solution, the time-dependent part is $\cos(\sqrt{d}\pi c t)$, so the angular frequency of temporal oscillation is $\omega = \sqrt{d}\pi c$. Thus, the wave speed $c$ directly determines the frequency: for a fixed spatial dimension $d$, the frequency is proportional to $c$. In other words, $c$ can be viewed as a parameter controlling the temporal oscillation frequency—the larger the value of $c$, the faster the solution oscillates in time.

### B.3  Viscous Burgers

The Burgers equation is a fundamental nonlinear PDE combining advection and diffusion, used as a prototype for shock formation and turbulence modeling. We consider the following Viscous Burgers:

$$\begin{cases} u_t + u\, u_x = \nu u_{xx}, \quad x \in [-1,1], \ t > 0, \\ u(-1,t) = 0, \ u(1,t) = 0, \quad t \geq 0, \\ u(x,0) = -\sin(\pi x), \quad x \in [-1,1]. \end{cases} \tag{B.7}$$

where viscosity $\nu = \frac{0.01}{\pi}$. With such small viscosity, the solution behaves almost inviscid: gradients steepen rapidly and form very thin viscous layers (shock transitions). Similarly, we set $T = 1$ and $t \in [0,1]$. Analytical solution is introduced in (Basdevant et al., 1986).

### B.4  Advection-Reaction-Diffusion

The Advection-Reaction-Diffusion (ARD) equation considered in our experiments takes the following general form:

$$\begin{cases} u_t + V_{\text{adv}} u_x - D_{\text{diff}} u_{xx} - \lambda(x)\big(u - u_{\text{eq}}(x)\big) = S(t,x), \quad x \in (0,1), \ t \in (0,1], \\ u(t,0) = u_{\text{exact}}(t,0), \quad t \in (0,1], \\ \partial_x u(t,1) = \partial_x u_{\text{exact}}(t,1), \quad t \in (0,1], \\ u(0,x) = u_{\text{exact}}(0,x), \quad x \in [0,1]. \end{cases} \tag{B.8}$$

where $V_{\text{adv}} = 1.0$ is the advection speed, $D_{\text{diff}} = 0.01$ the diffusion coefficient, $\lambda(x)$ a spatially varying reaction rate, $u_{\text{eq}}(x)$ an equilibrium profile, and $S(t,x)$ a source term. Note that a Dirichlet boundary

condition is applied at the left boundary ($x = 0$) and a Neumann boundary condition at the right boundary ($x = 1$).

The spatially varying reaction rate is defined as:

$$\lambda(x) = \lambda_0 + \frac{\Delta\lambda}{2}\Big[\tanh\big(\alpha(x - x_1)\big) - \tanh\big(\alpha(x - x_2)\big)\Big], \tag{B.9}$$

with $\lambda_0 = 0.02$, $\Delta\lambda = 20.0$, $\alpha = 50.0$, $x_1 = 0.45$, and $x_2 = 0.55$. This creates a smooth "bump" in the reaction rate over the interval $[x_1, x_2]$. The equilibrium profile is:

$$u_{\text{eq}}(x) = \frac{1}{2}\Big[1 + \tanh\big(\beta(x - 0.5)\big)\Big], \tag{B.10}$$

with $\beta = 40.0$, representing a smooth transition from 0 to 1 centered at $x = 0.5$.

To ensure the PDE holds exactly, the source term $S(t, x)$ in each case is explicitly constructed by analytically substituting a prescribed exact solution $u_{\text{exact}}(t, x)$ into the left-hand side of the equation.

**Case 1: Traveling Wave.** In this scenario, the exact solution represents a traveling wave propagating to the right with speed $V_{\text{adv}}$:

$$u_{\text{exact}}(t, x) = m - a \tanh\left(\frac{x - x_0 - V_{\text{adv}}t}{\varepsilon}\right), \tag{B.11}$$

where $m = \frac{U_L + U_R}{2}$ and $a = \frac{U_L - U_R}{2}$, with the left and right states given by $U_L = 2.0$ and $U_R = 1.0$. The initial interface is located at $x_0 = 0.35$, and the interface thickness is controlled by $\varepsilon = 0.02$.

**Case 2: Oscillatory Decay.** In this scenario, the exact solution is decoupled into a time-dependent amplitude and a stationary spatial front:

$$u_{\text{exact}}(t, x) = m - aA(t) \tanh\left(\frac{x - x_0}{\varepsilon}\right), \tag{B.12}$$

where the spatial parameters $m, a, U_L, U_R$, and $x_0$ remain identical to the traveling wave case, but with a sharper interface thickness $\varepsilon = 0.01$. The time-dependent amplitude is defined as:

$$A(t) = e^{-\gamma t} \cos(\omega t), \tag{B.13}$$

introducing an oscillatory decay with decay rate $\gamma = 0.2$ and angular frequency $\omega = 2\pi$. This creates a front that oscillates in time while its sharpness remains constant. Both cases present challenging benchmarks for assessing the model's ability to capture sharp fronts and spatially varying coefficients.

## B.5 Allen-Cahn

The Allen–Cahn equation is a well-known model for phase separation and interface dynamics. In this work, we consider the one-dimensional form with periodic boundary conditions:

$$\begin{cases} u_t = \gamma_1 u_{xx} - \gamma_2\big(u^3 - u\big), & x \in [-1, 1],\ t \in (0, 1], \\ u(0, x) = x^2 \cos(\pi x), & x \in [-1, 1], \\ u(t, -1) = u(t, 1), & t \in (0, 1], \\ \partial_x u(t, -1) = \partial_x u(t, 1), & t \in (0, 1]. \end{cases} \tag{B.14}$$

where $\gamma_1$ and $\gamma_2$ are positive constants controlling the diffusion strength and the depth of the double-well potential, respectively. In our experiments, we set $\gamma_1 = 10^{-3}$ and $\gamma_2 = 5.0$. These values yield a system where diffusion is relatively weak and the nonlinear reaction term dominates, leading to the rapid formation of steep phase boundaries that move and eventually saturate at the steady states $u \approx \pm 1$.

Because no closed-form analytical solution is available for this setup, we generate a high-fidelity reference solution using a Fourier pseudo-spectral method with implicit-explicit (IMEX) time stepping. The spatial

domain $[-1, 1)$ is discretized uniformly with $N = 2048$ points, and the time step is set to $\Delta t = 1/200$, yielding 200 integration steps up to the final time $T = 1.0$. Within this numerical scheme, the linear diffusion term is treated implicitly in Fourier space to ensure stability, while the cubic nonlinearity is handled explicitly. The resulting discrete solution, denoted as $u_{\text{ref}}(t_i, x_j)$, is stored on the full space-time grid to serve as the ground truth for model training, error evaluation, and visualization.

## C  Training Details

**Data Sampling and Preprocessing**  For each PDE, we sample $N_f$ collocation points in the interior domain, $N_{bc}$ points on the spatial boundaries, and $N_{ic}$ points at the initial time $t = 0$ using a **uniform distribution**. The specific counts for each case are summarized in Table 10. For the **Allen-Cahn** equation, we utilize the full spatio-temporal grid ($2048 \times 200$) from the Fourier pseudo-spectral reference solution to ensure sufficient supervision for the sharp interface.

To ensure the robustness of our results, all experiments are conducted across five random seeds: $\{1234, 42, 4, 2, 2025\}$. The results reported in the main text use **seed 1234** as the default.

Table 10: Experimental configurations and hyperparameters for all PDE benchmarks. $N/A$ indicates a component is not applicable to that specific problem.

| PDE Case | Points $(N_f, N_b, N_{ic})$ | Adam LR | Weights $(w_f, w_{bc}, w_{ic})$ |
|---|---|---|---|
| Viscous Burgers | $(10000, 200, 256)$ | $2 \times 10^{-3}$ | $(1, 1, 10)$ |
| Poisson (5D) | $(8192, 2048, N/A)$ | $5 \times 10^{-4}$ | $(1, 5000, N/A)$ |
| Wave Equation | $(8192, 1024, 1024)$ | $1 \times 10^{-3}$ | $(1, 100, 100)$ |
| Allen-Cahn | $(2048 \times 200)^{\dagger}$ | $1 \times 10^{-3}$ | $(1, 1, 1)$ |
| ARD (Traveling) | $(8192, 1024, 1024)$ | $1 \times 10^{-3}$ | $(1, 1, 1)$ |
| ARD (Oscillatory) | $(8192, 1024, 1024)$ | $1 \times 10^{-3}$ | $(1, 1, 1)$ |
| 5d nonlinear Poisson | $(8192, 2048, N/A)$ | $5 \times 10^{-4}$ | $(1, 5000, N/A)$ |

$^{\dagger}$ *The Allen-Cahn case utilizes the complete reference grid points for training.*

**Loss Function**  We follow the PINN framework to optimize the network parameters $\theta$. For a time-dependent PDE $\mathcal{F}(u) = 0$ with boundary conditions $\mathcal{B}[u] = 0$ and initial condition $u(x, 0) = u_0$, the total loss is defined as:

$$\mathcal{L}(\theta) = w_f \mathcal{L}_{\text{PDE}} + w_{bc} \mathcal{L}_{\text{BC}} + w_{ic} \mathcal{L}_{\text{IC}} + \lambda_{\text{gate}} \mathcal{L}_{\text{gate}} \tag{C.1}$$

where $\mathcal{L}_{\text{gate}}$ is the auxiliary balancing loss for our DomainMoE architecture to ensure balanced expert utilization. The individual residual components are defined as:

$$\mathcal{L}_{\text{PDE}} = \frac{1}{N_f} \sum_{i=1}^{N_f} \left\| \mathcal{F}(x_f^{(i)}, t_f^{(i)}; u_\theta) \right\|^2 \tag{C.2}$$

$$\mathcal{L}_{\text{BC}} = \frac{1}{N_{bc}} \sum_{j=1}^{N_{bc}} \left\| \mathcal{B}[u_\theta](x_{bc}^{(j)}, t_{bc}^{(j)}) \right\|^2 \tag{C.3}$$

$$\mathcal{L}_{\text{IC}} = \frac{1}{N_{ic}} \sum_{k=1}^{N_{ic}} \left\| u_\theta(x_{ic}^{(k)}, 0) - u_0(x_{ic}^{(k)}) \right\|^2 \tag{C.4}$$

All derivatives are computed via automatic differentiation. The specific scalar weights are listed in Table 10.

**Two-stage Optimization Strategy**  All benchmarks are trained using a rigorous two-stage optimization protocol:

- **Stage I: Adam Pre-training.** We utilize the Adam optimizer for $n_{\text{Adam}} = 10,000$ iterations. A **linear weight annealing** strategy is applied to the PDE residual $w_f$, increasing it from $w_f^{\text{init}} = 0.01$

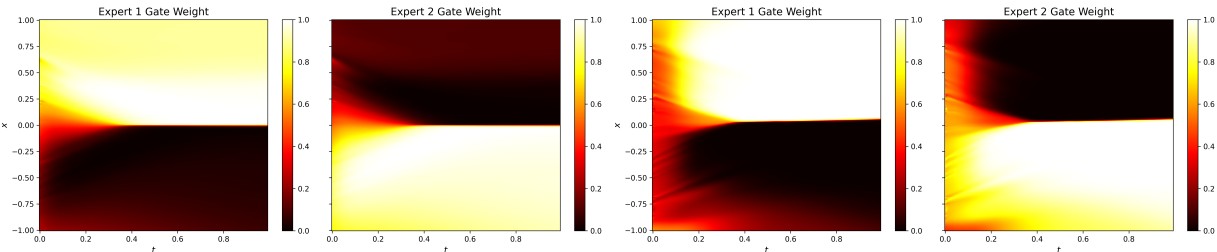

Figure 15: Robustness evaluation under boundary and initial condition corruption for the Viscous Burgers'
equation. (Left two) under a moderate noise level ($\sigma = 0.05$). (Right two) under an extreme noise level
($\sigma = 0.5$).

to $w_f^{\text{final}} = 1.0$ during the first $7,500$ steps (75% of the Adam stage). This curriculum learning
approach prioritizes the fulfillment of boundary and initial conditions before fitting the complex
PDE residuals. The learning rate follows a cosine annealing schedule from its initial value down to
$10^{-6}$.

- **Stage II: L-BFGS Refinement.** After the Adam stage, the model is further optimized using the
  L-BFGS algorithm (`torch.optim.LBFGS`) to reach high-precision convergence. We set the maximum
  number of iterations to $20,000$, with a history size of $100$ and the *Strong-Wolfe* line search. The
  optimizer terminates early if the change in loss is less than $10^{-12}$ or the gradient norm falls below
  $10^{-9}$.

## D  Other experiments

### D.1  Structural Robustness of Learned Domain Decomposition

To evaluate the robustness of our 3D framework on domain decomposition, we add zero-mean Gaussian
noise $\epsilon \sim \mathcal{N}(0, \sigma^2)$ on the boundaries and initial conditions across PDE benchmarks. See Figures 15, 16
and Table 11 for details. Errors are averaged across five random seeds. We observe that under a **moderate
noise level** ($\sigma = 0.05$), the learned domain decomposition across all systems remain virtually unchanged,
and the relative $\ell_2$ errors degrade.

Table 11: Relative $\ell_2$ error under noises.

| Problem | Burgers ($\sigma = 0.05$) | Burgers ($\sigma = 0.5$) | ADR (Wave)($\sigma = 0.5$) | ADR (Decay)($\sigma = 0.5$) |
|---|---|---|---|---|
| **Error** | $1.43 \times 10^{-2}$ | $3.85 \times 10^{-1}$ | $9.46 \times 10^{-3}$ | $5.59 \times 10^{-3}$ |

We further conduct an **extreme stress with a noise level of** $\sigma = 0.5$. For Viscous Burgers, even when
such severe noise causes the background solution landscape to be chaotic, the router still isolates the sharp
shock. For ADRs with distinct structural patterns, even this level of noise does not affect the domain
decomposition results. This structural preservation under extreme noise demonstrates that our framework
actively captures the intrinsic geometric invariants.

### D.2  2d Poisson with L-shape domain

To further show the generality of our framework on the irregular domain, we test our method on the 2d
Poisson on an L-shaped domain following the settings used in SPINNs (Cho et al., 2023):

$$\begin{cases} -\Delta u(x,y) = 1 & \mathbf{x} \in \Omega, \\ u(x,y) = 0 & \mathbf{x} \in \partial\Omega. \end{cases} \tag{D.1}$$

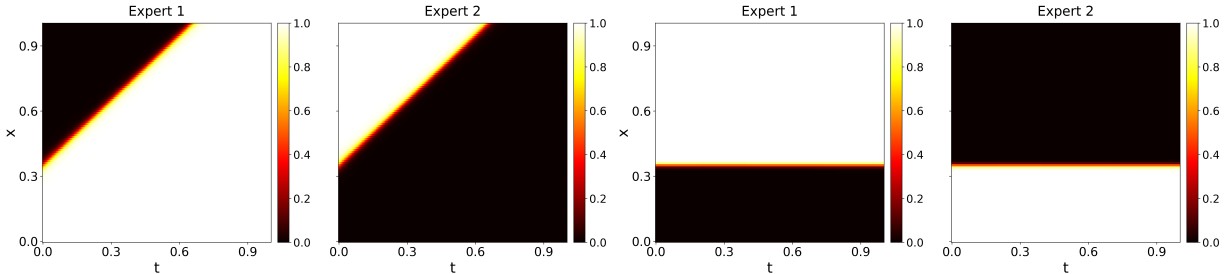

Figure 16: Robustness evaluation for the ADR equations under noise level $\sigma = 0.5$. (Left two) Traveling Wave. (Right two) Oscillatory Decay.

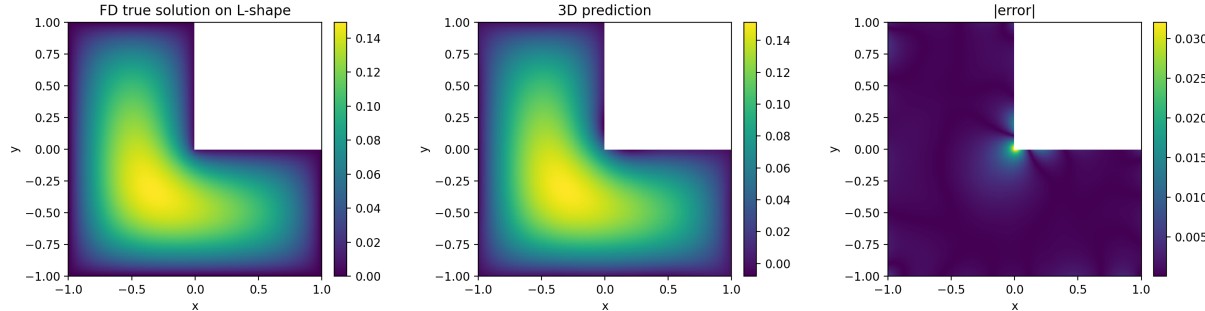

Figure 17: Ground truth, predicted solution and absolute error plots for 2d Poisson with L-shape domain.

where $\Omega = [-1, 1]^2 \setminus [0, 1]^2$. We use high-accuracy finite difference method for solving reference true solutions.

Specifically, only $N_{\text{int}} = 10{,}000$ interior collocation points are drawn from the two disjoint rectangles $[-1, 1] \times [-1, 0]$ and $[-1, 0] \times [0, 1]$, using power-law sampling ($|x| \sim U[0, 1]^\beta$, $\beta = 2.5$) to concentrate points near the re-entrant corner. Boundary points are generated uniformly along the polygonal boundary of the L-shape with $N_{\text{bc}} = 200$ points per edge. The model uses a shared MLP with four hidden layers of width 64 and rank $r = 32$. Under this setting, our model achieves the relative $\ell_2$ error of $2.5520 \times 10^{-2}$ while SPINNs achieves the relative $\ell_2$ error of $2.9121 \times 10^{-2}$. The result is shown in Figure 17.

## D.3 Generalization to Irregular Domains with Non-Cartesian Boundaries

We conduct an additional experiment on an irregular, non-convex annular domain. This setup explicitly tests the router's capacity to naturally learn non-linear, curved interior segmentations without explicit geometric priors or global coordinate transformations.

We consider a 2d nonlinear Poisson equation defined on an annular domain $\Omega = \{(x, y) \mid R_{\text{in}} \leq \sqrt{x^2 + y^2} \leq R_{\text{out}}\}$, with $R_{\text{in}} = 0.2$ and $R_{\text{out}} = 1.0$:

$$\begin{cases} -\Delta u(x, y) + \lambda u^3(x, y) = f(x, y), & (x, y) \in \Omega, \\ u(x, y) = g(x, y), & (x, y) \in \partial\Omega, \end{cases} \tag{D.2}$$

where $\lambda = 1.0$. To simulate a highly localized and challenging solution structure that does not align with traditional rectangular grids, we construct an exact solution featuring a sharp, circular shock front modulated by angular oscillations:

$$u_{\text{true}}(x, y) = \cos(2\theta) \cdot \tanh\left(\frac{R_0 - r}{\epsilon}\right), \tag{D.3}$$

where $r = \sqrt{x^2 + y^2}$, $\theta = \arctan 2(y, x)$, $R_0 = 0.6$ represents the radial position of the shock front, and $\epsilon = 0.04$ controls the steepness of the shock. The forcing term $f(x, y)$ and the Dirichlet boundary condition $g(x, y)$ are analytically derived from $u_{\text{true}}$.

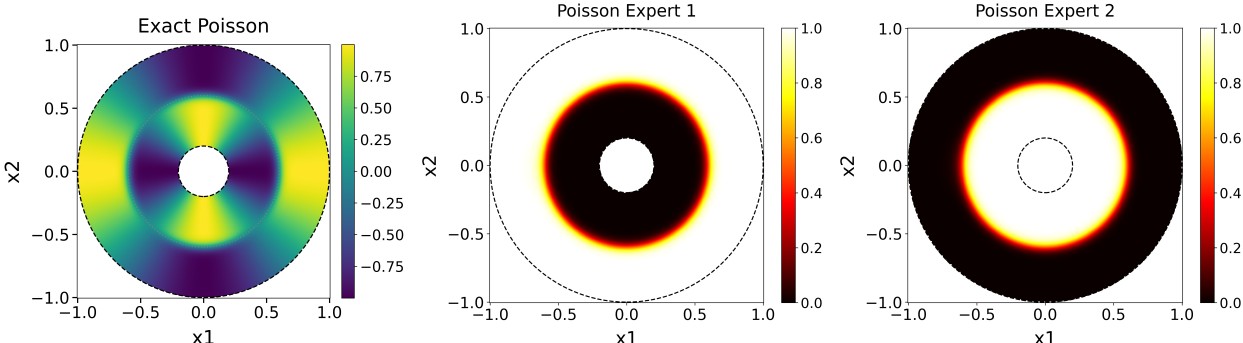

Figure 18: Numerical solution and learned domain decomposition for the 2d nonlinear Poisson equation with irregular domains with $K = 2$ experts.

We employ 3D framework configured with rank $r = 4$ and $K = 2$ experts. Crucially, the gating network receives only raw Cartesian coordinates $(x, y)$ as inputs. The domain is sampled using $N_{\text{int}} = 10,000$ interior points and $N_{\text{bc}} = 2,000$ boundary points distributed uniformly across the concentric boundaries.

As shown in Figure 18, the 3D framework tracks the circular, oscillating solution structure with high fidelity, achieving the relative $\ell_2$ error $1.5337 \times 10^{-3}$. Although the inputs are simple linear coordinates $(x, y)$, the gating network cleanly maps a concentric, non-linear division precisely at the shock radius $r \approx 0.6$. This result directly confirms that our soft MoE routing does not suffer from Cartesian limitations.

## D.4 Cross-Dimensional Transfer on Physical Green's Functions

Under 3D framework, fine-tuning is not restricted to simple case. We further investigate the framework's capability to transfer learned representations across spaces governed by distinct physical and singularity structures. We consider the multi-dimensional Poisson equation with a singular Dirac delta source at the origin, whose solution yields the classic Green's function. The solution $u(\mathbf{x})$ depends strictly on the radial distance $r = \|\mathbf{x}\|_2$. As dimension changes, the underlying physical field and mathematical singularity shift fundamentally:

$$
\begin{cases}
u_{2\text{D}}(\mathbf{x}) = -\dfrac{1}{2\pi}\ln(r), & \mathbf{x} \in \mathbb{R}^2 \setminus \{\mathbf{0}\}, \\
u_{3\text{D}}(\mathbf{x}) = \dfrac{1}{4\pi}\dfrac{1}{r}, & \mathbf{x} \in \mathbb{R}^3 \setminus \{\mathbf{0}\}.
\end{cases}
\tag{D.4}
$$

The 2d field exhibits a slow logarithmic singularity that diverges at infinity, whereas the 3d field exhibits a sharp algebraic singularity $(\frac{1}{r})$ that decays rapidly. This means that transferring a model from 2d to 3d requires the neural network to completely restructure its internal feature mappings rather than merely appending an extra coordinate axis. The empirical profiles across both cases are detailed in Figure 20. Training dynamics and numerical results are shown in Figure 19 and Table 12. This confirms that our cross-dimensional transfer capability stands strong even under physics shifts.

## D.5 Multi-Region Domain Decomposition

We also evaluate the 3D on a non-homogeneous stationary advection-diffusion-reaction (ADR) equation involving complex multi-zone physical behaviors. Specifically, we consider the **steady-state multi-physics transport equation** defined on the domain $\Omega = [-1, 1]^2$:

$$
\mathbf{b}(x, y) \cdot \nabla u(x, y) + \sigma(x, y)u(x, y) = f(x, y), \quad (x, y) \in \Omega
\tag{D.5}
$$

subject to the Dirichlet boundary condition $u(x, y) = u_{\text{exact}}(x, y)$ on $\partial\Omega$.

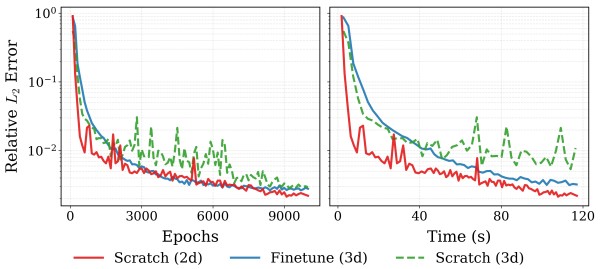

Figure 19: Convergence dynamics.

Table 12: Performance comparison across model variants.

| Model | Epochs | Time (s) | Error ($\times 10^{-4}$) |
|---|---|---|---|
| 2d (src) | 13200 | 161.29 | $7.26 \times 10^{-4}$ |
| 3d (src) | 14000 | 343.84 | $5.81 \times 10^{-4}$ |
| 3d (fine) | 14300 | 361.66 | $5.69 \times 10^{-4}$ |

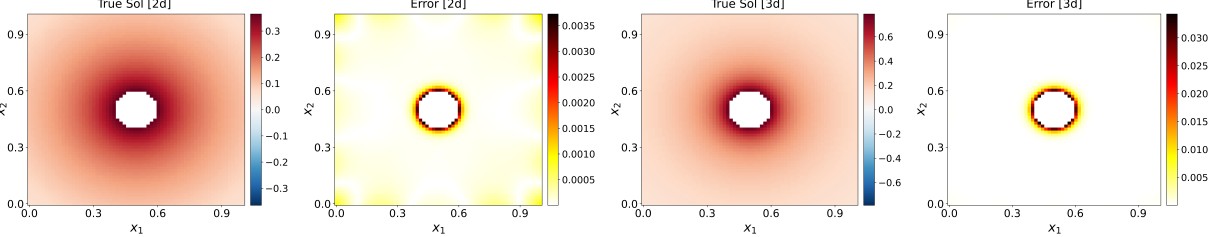

Figure 20: True solution and pointwise absolute error for 2d and fine-tuned 3d nonlinear Poisson equation.

The spatial domain is sharply segmented into three distinct vertical subdomains $(\Omega_1, \Omega_2, \Omega_3)$ via latent, unaligned interfaces located at $x_1 = -1/3$ and $x_2 = 1/3$. The material properties and transport mechanics are governed by piecewise constant coefficient fields:

$$\mathbf{b}(x,y) = \begin{cases} (3.0, 1.0)^T, & x < -1/3 \quad (\Omega_1 : \text{Fast Advection}) \\ (0.1, 0.1)^T, & -1/3 \leq x \leq 1/3 \quad (\Omega_2 : \text{Strong Reaction}) \\ (1.0, -2.0)^T, & x > 1/3 \quad (\Omega_3 : \text{Mixed Advection}) \end{cases} \tag{D.6}$$

$$\sigma(x,y) = \begin{cases} 0.1, & x < -1/3 \\ 10.0, & -1/3 \leq x \leq 1/3 \\ 1.0, & x > 1/3 \end{cases} \tag{D.7}$$

The non-homogeneous source term $f(x,y)$ is analytically manufactured using a synthetic exact solution $u_{\text{exact}}(x,y)$.

$$u_{\text{exact}}(x,y) = w_1(x)u_1(x,y) + w_2(x)u_2(x,y) + w_3(x)u_3(x,y) \tag{D.8}$$

where the sub-solutions within each independent domain are defined as:

$$u_1(x,y) = \sin(\pi(x+1.0))\sin(\pi y) \tag{D.9}$$

$$u_2(x,y) = \exp(-5.0x^2)\sin(\pi y) \tag{D.10}$$

$$u_3(x,y) = \sin(2\pi(x-x_2))\sin(\pi y) \tag{D.11}$$

The space-dependent blending weights $w_1(x), w_2(x)$, and $w_3(x)$ are governed by smooth step functions localized at the latent interfaces $x_1 = -1/3$ and $x_2 = 1/3$:

$$w_1(x) = 1 - \mathcal{S}(x - x_1) \tag{D.12}$$

$$w_2(x) = \mathcal{S}(x - x_1) - \mathcal{S}(x - x_2) \tag{D.13}$$

$$w_3(x) = \mathcal{S}(x - x_2) \tag{D.14}$$

here, $\mathcal{S}(t) = \frac{1}{1+\exp(-kt)}$ represents a sharp parameterized sigmoid function with a steepness scaling factor of $k = 40.0$. This formulation ensures that the exact solution transitions rapidly yet continuously across boundaries, presenting stringent requirements for local gradient approximations.

As demonstrated in Figure 21, the router automatically discovers the latent discontinuities. 3D captures the distinct physics with a relative $\ell_2$ error of $9.96 \times 10^{-3}$.

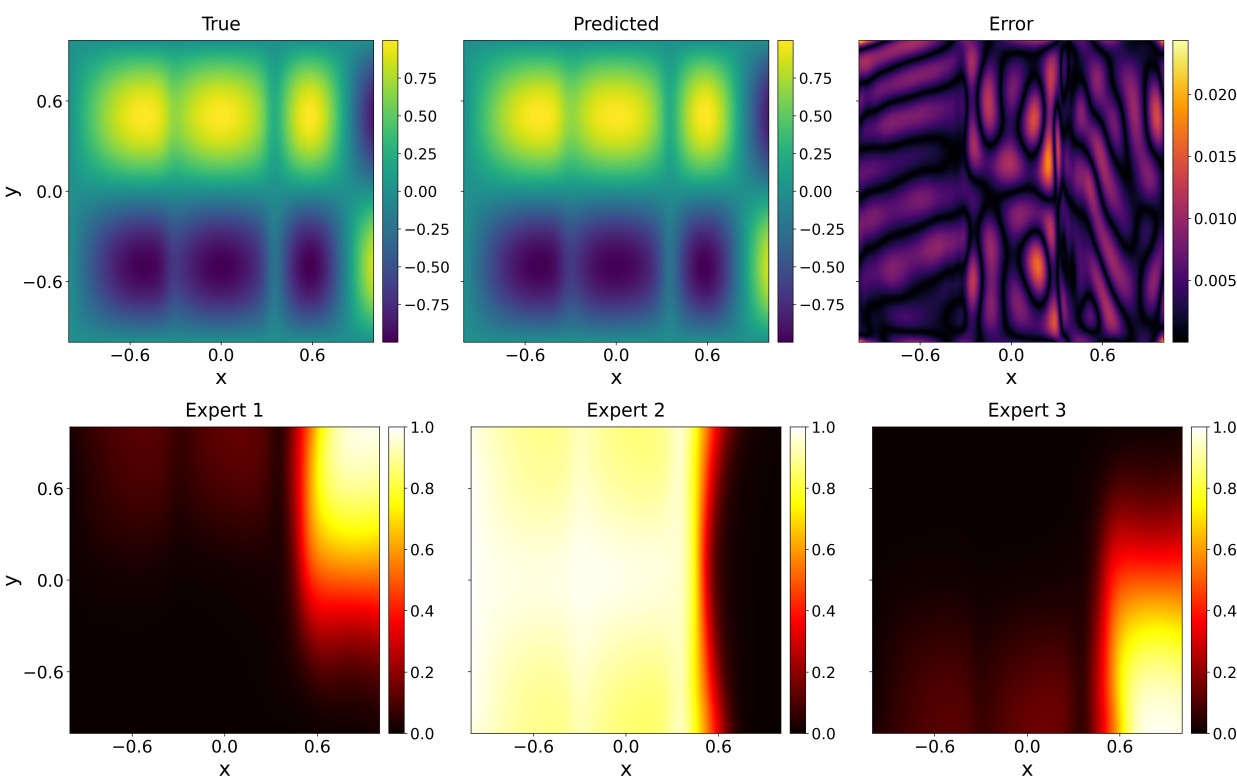

Figure 21: Solution profiles and domain decomposition results for the transport equation with $K = 3$. First row: true solution, predicted solution and error plot. Second row: gates assignment of each expert.

