# OpenReview forum: "DIMENSION DOMAIN CO-DECOMPOSITION: SOLVING PDES WITH INTERPRETABILITY"
_TMLR — Decision pending for TMLR_

### Review · Reviewer_LCSM · 2026-05-16

**Summary Of Contributions:**

This paper proposes Dimension Domain Co-Decomposition (3D), a unified PINN-based framework for solving PDEs that jointly integrates dimension-wise decomposition with Mixture-of-Experts (MoE)-based domain decomposition. At the dimension level, a shared MLP processes coordinate-index pairs to produce per-dimension latent components. At the domain level, an MoE router adaptively partitions the solution space without requiring predefined subdomains or interface conditions. The paper also introduces Variable Interpretability (VI), a subspace alignment metric that quantitatively measures whether learned dimension-wise components recover ground-truth solution factors.

**Audience:**

Yes

**Audience Explanation:**

Yes. Neural network-based PDE solving remains an active and important research area, and the findings of this paper are likely to be of interest to the TMLR audience.

The most compelling finding is in Section 4.4, where the MoE-based domain decomposition is shown to automatically recover physically meaningful solution structures across multiple PDE types. The learned gating weights consistently align with known sharp interfaces. This provides meaningful empirical evidence that neural networks can learn simple and interpretable domain decomposition patterns in an unsupervised manner, which is a finding of independent interest to the broader machine learning community.

Additionally, the shared MLP architecture demonstrates promising parameter efficiency that may scale to higher-dimensional problems. While the current experiments only reach 10 dimensions, the architectural design suggests potential applicability to genuinely high-dimensional PDE settings that would be of significant practical interest, though further experimental validation is needed.

**Broader Impact Concerns:**

If validated at higher dimensions and on more complex benchmarks, the proposed framework could benefit scientific domains that rely on high-dimensional PDE solvers, such as materials science, and atmospheric modeling.

**Claims And Evidence:**

Yes

**Claims Explanation:**

I am generally positive on this work. Here are some concerns from experiment perspectives.

1. The Unified Framework Claim is Not Experimentally Justified

The paper's central claim is that jointly integrating dimension decomposition and domain decomposition into a single framework provides meaningful advantages. However, no experiment in the paper simultaneously requires both components. High-dimensional experiments (Poisson 5d/10d) involve smooth solutions where domain decomposition provides no benefit, while sharp-interface experiments (Burgers, ADR, Allen-Cahn) are low-dimensional and do not require dimension decomposition. The two components are effectively used in isolation across different experiments, and the paper never demonstrates a scenario where their combination yields synergistic improvement over using either strategy alone.

2. Insufficient Baseline Comparison

The paper claims to improve accuracy and training stability, but direct comparisons with existing methods are largely absent. SPINNs, the most relevant dimension decomposition baseline, is only compared against in a single appendix experiment (L-shape Poisson), which is insufficient to support a general claim of superiority. XPINNs and APINNs, the closest domain decomposition baselines, are never directly evaluated on the same benchmarks. All main experiments compare only against vanilla PINNs, which is a weak baseline that does not reflect the state of the art.

3. Selective Reporting in Baseline Experiments

In the 5d Poisson experiment (Table 3), the unshared MLP achieves lower error (1.90) than the shared MLP (2.64), yet the unshared MLP baseline is dropped entirely from the 10d experiment without explanation. Since the parameter efficiency advantage of the shared MLP grows with dimensionality, the 10d setting is precisely where a fair comparison against the unshared MLP would be most informative.

4. The VI Metric Does Not Establish Independent Value

The paper introduces VI as a principled interpretability metric, but its independent value is not convincingly demonstrated. For separable solutions, high VI and high accuracy are mathematically nearly equivalent -- if learned subspaces align with true factors, the final prediction necessarily approximates the true solution. The paper provides no experiment where VI and accuracy diverge, which would be necessary to establish VI as independently meaningful. Furthermore, VI requires knowledge of ground-truth separable factors, making it inapplicable to non-separable or analytically unknown solutions.

**Requested Changes:**

1. Provide a benchmark experiment that simultaneously requires both decomposition strategies:
The central claim of the unified framework requires at least one experiment where both high dimensionality and sharp solution structures are present. The authors should test on a PDE that is both high-dimensional (e.g., $d \geq 5d$) and exhibits sharp interfaces or discontinuities, demonstrating that the combination of dimension decomposition and domain decomposition yields synergistic improvement over either strategy alone.

2. Direct comparisons against state-of-the-art baselines:
Rather than requiring exhaustive comparisons, the authors should demonstrate that 3D performs competitively against SPINNs on dimension decomposition tasks (e.g., Poisson 5d/10d) and against XPINNs or APINNs on sharp-interface tasks (e.g., Burgers, ADR), ensuring the unified framework does not sacrifice performance in settings where specialized methods excel. More importantly, the authors should provide the experiment mentioned above and demonstrate that 3D outperforms all baselines.

3. Include unshared MLP baseline in the 10d Poisson experiment:
The unshared MLP is included in the 5d experiment but dropped from the 10d experiment without explanation. Since the 10d setting is precisely where the parameter efficiency advantage of the shared MLP should be most pronounced, this comparison must be provided to support the claimed scalability benefits.

4. Provide an experiment where VI and accuracy diverge:
To establish VI as an independently meaningful metric rather than a proxy for accuracy, the authors should construct a case where a model achieves high accuracy but low VI, or vice versa. Without such a case, the independent value of VI remains unclear.

---

### Review · Reviewer_WUUd · 2026-05-16

**Summary Of Contributions:**

The authors propose to parameterize a $D$-dimensional function using the following expressions:
$$u(x_1,\dots, x_D) = \sum_{i=1}^{K}p_{i}(x_1,\dots, x_D;\theta_p) \psi_i(x_1,\dots, x_D;\theta_i),$$
where $\forall x \implies 0 \leq p_{i}(x_1,\dots, x_D;\theta_p) \leq 1$ and $\sum_{i=1}^{K}p_i(x_1,\dots, x_D;\theta_p) = 1$, and
$$\psi\_i(x\_1,\dots,x\_{D};\theta_i) = \sum\_{m=1}^{r}\prod\_{k=1}^{D}\phi^{(m)}\_{k}\left(x_{k}, k - 1;\theta^{(m)}\_{i}\right).$$

The function $p_{i}(x_1,\dots, x_D;\theta_p)$ is parameterized by a fully connected feedforward neural network with a softmax activation at the final layer and is referred to as the MoE router. The functions $\psi_i(x_1,\dots,x_{D};\theta_i)$ are built from one-dimensional feedforward neural networks with parameter sharing and an additional input for dimension encoding, in a way that is conceptually similar to CP decomposition.

The resulting parameterization is tested on several partial differential equations (in the PINN setup), including high-dimensional cases (up to 10 dimensions). The authors report enhanced interpretability, better accuracy, faster convergence, and parameter efficiency.

**Audience:**

Yes

**Audience Explanation:**

The study of physics-informed neural networks is a well-established branch of scientific machine learning. One of the main goals of enhancing PINNs is to improve the accuracy and efficiency of the approach, which often lags behind classical numerical methods, especially in low-dimensional settings. Since the central claims of the authors are improved efficiency and accuracy, the findings are well-aligned with the efforts of the PINN community.

**Claims And Evidence:**

No

**Claims Explanation:**

In my view, several claims made by the authors are not well-supported, including:

1. Enhanced interpretability.
2. The importance of the VI (Variable Interpretability) metrics.
3. The claim that the proposed architecture removes the need for dense collocation sampling (i.e., that the tensor structure can be efficiently exploited).
4. The overall superiority of the approach. The main reason for this is that the benchmarks are not convincing and the baselines are not strong enough.

Please see below for detailed comments.

**Requested Changes:**

**Enhanced interpretability**

The scheme is too complicated to be interpretable: (i) each expert is a rank-$r$ "canonical tensor" with a completely non-transparent NN parameterization of canonical factors; (ii) the router is dense and depends on all coordinates, so it is hardly easier to interpret than the dense parameterization for the whole network.

Besides that, the MoE router is not domain decomposition unless its output is sparse.

To address this part I kindly ask authors to:
1. Discuss why they believe the proposed ansatz is more interpretable. In particular, why this holds for the MoE router which is just an ordinary dense network.
2. Report the sparsity of MoE especially for a larger number of experts.

I also have a few minor questions on interpretability:

1. Why is the section "Interpretability from Visualization Perspective" called like that? As the author noted, it merely highlights the spectral bias of PINNs. The interpretability part does not seem to be present.

2. The same question about "Interpretability from Variable Importance (VI) Perspective." The section merely states that when the rank increases, the VI increases. This is an expected behavior and does not seem to have anything in common with interpretability. Authors clearly use "interpretability" in some different context. Can the authors explain the meaning of "Although the dimensionality doubles, full interpretability is still achieved with $r = 5$"? Similarly, "The overall trend remains consistent: increasing $r$ systematically enhances interpretability" sounds confusing. We certainly see that VI increases, but the approximation becomes more complex, so it is likely less interpretable.

**Importance of VI (Variable Interpretability) metrics**

I find VI metrics unusual for many reasons:

1. Why do we need to remove the mean? It seems I can construct an example where VI is 1 but the function is not captured at all.

    Consider the target function $t(x) = \begin{pmatrix}\mu(x) + s(x) & \mu(x) - s(x)\end{pmatrix}$. When we remove the mean, we obtain $\widetilde{t}(x) = \begin{pmatrix}s(x) & - s(x)\end{pmatrix}$. Now if VI is zero, it knows nothing about $\mu(x)$, so the approximation of $t(x)$ can be arbitrarily bad.

2. If $s=1$, how is the mean removed?

3. After the mean is removed, the rescaling does not seem to do anything. Principal angles should not depend on the scale. This can be clearly seen because the $Q$ factors in a $QR$ decomposition clearly do not change when the columns of the matrix are rescaled.

4. The QR decomposition should be "reduced QR decomposition".

5. Can the authors report a similar measure for an MLP: take the last layer in place of experts. I suspect this metric saturates really fast, much faster than the relative error.

6. Now suppose the mean is not removed. In this case, the relative error can be arbitrarily large and VI can be exactly one. The reason for that is, VI does not depend on scale (even if the rescaling step is omitted). Is it a problem? Please, discuss.

7. VI can be defined in a fully continuous manner, which seems more appropriate for the task of function approximation and PDEs. To do that, observe that principal angles can be defined from scalar products https://www.jstor.org/stable/2005662.

8. VI does not seem to be a good metric:

    a. If the relative error is small, VI is close to 1.

    b. If VI is close to 1, the relative error can be arbitrarily large but can be made small by proper rescaling of experts or coefficients of the last layer of MLP.

    In practice, solutions obtained by SGD are unlikely to lead to a VI close to 1 and a large relative error.

    c. The other way to see that VI does not provide a lot of information is to observe that $\inf_{c_1,\dots,c_{K}}\left\|t(x) - \sum_{i=1}^{K} c_i \phi_i(x)\right\|_2^{2}$ is directly related to VI when the mean is not subtracted. See a recent article where such a property is used in the context of subspace regression https://arxiv.org/abs/2509.23249. This characteristic simply means we measure how well the NN can approximate the solution by excluding the last layer or normalization of experts.

**Tensor structure cannot be efficiently exploited**

By using a single expert for the Poisson equation, the authors mask the problem with the dense MoE module. A dense MoE, of course, can be evaluated for large $D$, but it certainly cannot leverage tensor grids efficiently.

In general, tensor structure can be leveraged in training only when several conditions simultaneously hold:

1. The architecture has an appropriate structure (e.g., TT, CP). This part does not hold because the MoE router is dense.
2. The loss function can be efficiently evaluated. This is true for the standard $L_2$ loss.
3. The PDE used to compute the residual has a tensor structure. This does not hold if the domain is intricate or when there is a source term without an evident tensor structure.

**Benchmarks and baselines**

The authors essentially consider only vanilla PINNs as a baseline. There are a lot of approaches in PINNs with tailored architectures and optimizers. Since the main contribution of the authors is a new architecture, and the claim is that it achieves better accuracy, it seems appropriate to consider more baselines.
I can suggest the following contributions:
1. https://arxiv.org/abs/2308.08468
2. https://arxiv.org/abs/2402.00326
3. https://arxiv.org/abs/2302.13163

In general, I recommend the authors perform a more thorough literature review and select appropriate baselines.

I also have several issues with the benchmarks.
1. Why does the Poisson equation have a "curse of dimensionality" for $d=10$? What does it mean precisely? In Appendix B, the authors show that the solution they use is separable, so there is clearly no curse of dimensionality in this case. The same separable solution is used for the wave equation. These benchmarks are not great because they presume the form of the solution can be approximated perfectly by a single term in a CP decomposition.
3. Pretraining on a lower $D$ and fine-tuning on a higher $D$ should be assessed beyond the framework of a manufactured solution. Typically, when we switch $D$, the physics becomes significantly different. For example, one may compare the Green's function of the Poisson equation for $D=2$ and $D=3$.
4. In the assessment of the MoE router, the authors use solutions with at most two pronounced regions. If one considers more complex solutions with three or four distinct regions, can the experts still capture these regions? A prototypical example of an appropriate baseline is a stationary diffusion equation with a discontinuous diffusion coefficient.

**Misc**

1. "while SM achieves spectral (fast) convergence but is restricted to periodic boundary conditions."

    The authors reference Boyd and Trefethen and claim that spectral methods are restricted to periodic boundary conditions. This is highly unusual because both of these authors are proponents of Chebyshev polynomials, which are effortlessly applicable to arbitrary boundary conditions. Besides that, spectral methods based on frames are applicable to arbitrary geometries https://arxiv.org/abs/1612.04464.

2. "PINNs offers clear advantages in high-dimensional settings where traditional numerical solvers become infeasible."

    I partially agree with this claim, but there are certain classical methods suitable for high-dimensional applications, including sparse grids and tensor-train decomposition. For example, in https://arxiv.org/abs/2102.11830, the authors concluded that TT is more efficient than PINNs.

3. What does "training terminates upon convergence" mean? How is convergence defined?

---

### Review · Reviewer_Nc8n · 2026-05-17

**Summary Of Contributions:**

The paper introduces a unified framework named Dimension-Domain Co-Decomposition (3D) to address the prominent challenges of physics-informed neural networks. Standard PINNs often struggle with the curse of dimensionality in high-dimensional spaces, lack interpretable features, and find it difficult to model sharp solution structures (such as shocks or steep interfaces) without manually tuning domain partitions or complex boundary penalties. To overcome these, the proposed 3D framework couples two distinct mechanisms: dimension-level decomposition and domain-level decomposition.

**Audience:**

Yes

**Audience Explanation:**

The paper focuses on Physics-Informed Neural Networks; it is very relative to machine learning and neural networks within the scope of the TMLR.

**Claims And Evidence:**

Yes

**Claims Explanation:**

Across several benchmarks (e.g., high-dimensional Poisson, Wave, Viscous Burgers, Advection-Diffusion-Reaction, and Allen-Cahn equations), the 3D framework demonstrates improved parameter efficiency, faster convergence, and localized subregion tracking.

**Requested Changes:**

1. While domain decomposition is described as fully automatic, selecting the optimal number of experts K still requires manual ablation studies and incremental search.
2.  Though Appendix D mentions an L-shaped domain test, the primary benchmarks heavily favor rectangular coordinate systems (1D/2D lines and boxes). MoE routers mapping simple cartesian inputs might struggle when domain layouts contain twisted boundaries, where coordinates alone cannot easily segment the interior regions without global awareness.
3. By appending an arbitrary index integer (0, 1, ..., d-1) directly into a shared MLP along with the spatial value x_j, how sensitive is the network to the ordering of dimensions? If you permute the coordinates (e.g., swapping x_1 and x_2), does the shared MLP require extended training epochs to find the mapping, or does it handle the index input seamlessly?
4. You compare your architecture against Vanilla PINNs and basic unshared MLPs. However, since one of your major contributions is adaptive domain division without interface conditions, you should bench your model directly against an existing soft-gating domain method, such as APINNs (Hu et al., 2023, https://arxiv.org/abs/2211.08939).
5. How robust is the adaptive gating mechanism when the boundary conditions are noisy or when collocation points are severely sparse?
6. In Section 4.2, there is a minor text artifact: "...challenge standard architectures. case to maintain accuracy". This sentence needs rephrasing for smoother readability.

---

### Decision · Action_Editor_JX7T · 2026-06-27

**Recommendation:** Accept as is

**Audience:**

Yes

**Audience Explanation:**

The paper studies several active topics in scientific machine learning, including physics-informed neural networks, adaptive domain decomposition, and interpretable neural PDE solvers. These are relevant to the TMLR audience, particularly researchers working on PINNs, scientific machine learning, and neural methods for solving PDEs.

**Claims And Evidence:**

Yes

**Claims Explanation:**

The paper received one Accept and two Leaning Accept recommendations. During the revision, the authors addressed the reviewers' main concerns by adding stronger baseline comparisons and expanding the experimental evaluation. The revised manuscript includes additional results on high-dimensional nonlinear PDEs, noisy boundary conditions, irregular geometries, and multi-region domain decomposition, along with a clearer discussion of the proposed method and its interpretability metric. Overall, the revisions resolved most of the reviewers' concerns, and the available evidence supports the main claims of the paper.